# 👧 OBSCURACODER: POWERING EFFICIENT CODE LM PRE-TRAINING VIA OBFUSCATION GROUNDING

**Indraneil Paul**◎*✉ **Haoyi Yang**◑* **Goran Glavaš**⊡ **Kristian Kersting**◑ **Iryna Gurevych**◎
◎**UKP Lab**, TU Darmstadt ◑**AIML Lab**, TU Darmstadt ⊡**CAIDAS**, JMU Würzburg

## ABSTRACT

Language models (LMs) have become a staple of the code-writing toolbox. Their pre-training recipe has, however, remained stagnant over recent years, barring the occasional changes in data sourcing and filtering strategies. In particular, research exploring modifications to Code-LMs' pre-training objectives, geared towards improving data efficiency and better disentangling between syntax and semantics, has been noticeably sparse, especially compared with corresponding efforts in natural language LMs. In this work, we examine grounding on obfuscated code as a means of helping Code-LMs look beyond the surface-form syntax and enhance their pre-training sample efficiency. To this end, we compile **ObscuraX**, a dataset of approximately 55M source and obfuscated code pairs in seven languages. Subsequently, we pre-train **ObscuraCoder** models, ranging in size from 255M to 2.8B parameters, on a 272B-token corpus that includes ObscuraX and demonstrate that our obfuscation-based pre-training recipe leads to consistent improvements in Code-LMs' abilities compared to both vanilla autoregressive pre-training as well as existing de-obfuscation (DOBF) objectives. ObscuraCoder demonstrates sizeable gains across multiple tests of syntactic and semantic code understanding, along with improved capabilities in multilingual code completion, multilingual code commit summarization, and multi-purpose library-oriented code generation.

## 1 INTRODUCTION

In recent years, language models for code generation (Code-LMs) have become a near-indispensable accessory in a developer's toolbox. Their enhancement of productivity has proliferated into most of the software development lifecycle, including automating unit test generation, code infilling, predicting build errors, and code refactoring, *inter alia*, propelling their broad adoption in production (Dunay et al., 2024; Frömmgen et al., 2024; Murali et al., 2024). Code-LMs owe their competence gains in these areas to an ever-increasing parameter count (Liu et al., 2024) coupled with ever-larger scale of their pre-training: code corpora is not only increasing in size but also in quality due to progressively better data sourcing and filtering (Yang et al., 2024). While improvements to the hardware (Prabhakar et al., 2024) and software (Shah et al., 2024) stack are likely to continue and in turn facilitate training of even larger models, LM training scaling-laws (Hoffmann et al., 2022) predict diminishing gains from the relatively slow growth of publicly available code corpora (Lozhkov et al., 2024). This warrants a departure from the dominant trend of autoregressively training larger Code-LMs on more code and towards alternative pre-training objectives that more efficiently exploit the inherent structure and regularity (Hindle et al., 2016) of source code.

**Disentangling code syntax and semantics.** All modern high-level programming source code inherently contains two information channels: syntactic and semantic (Knuth, 1984; Casalnuovo et al., 2020). While the former refers to conventional syntactic and naming choices developers make to facilitate interpretation and maintainability of the code (Casalnuovo et al., 2019), the latter pertains to the algorithmic intent behind the code. These two channels latently constrain each other and cannot be (fully) separated in form, only in understanding. To become a mainstay of software development, Code-LMs must excel in both syntactic and semantic code understanding.

The mastery of this dual-channel nature of programming code has so far, however, eluded Code-LMs. Existing studies show that Code-LMs' grasp of syntax is much better and obtained much

---

\* Equal contribution
✉ Corresponding author: indraneil.paul@tu-darmstadt.de

quicker during pre-training than their understanding of code semantics (Naik et al., 2022), with their representations being sensitive to semantic-preserving surface-form changes (Rabin et al., 2021). The standard LM objective latches on to common syntactic sub-spans, failing to capture the higher-level constructs in code (Ma et al., 2022) or common modes of usage (Fried et al., 2023), shown to be efficiently captured by specialized architectures (Shi et al., 2022). This is especially true for verbose languages, where separators account for a significant portion of all tokens (Rahman et al., 2019).

Consequently, disentangling Code-LMs' understanding of code syntax and semantics can unlock improvements on downstream tasks. Evidence for this has been demonstrated at inference-time: Code-LMs completion performance drastically improves when provided with fine-grained syntactic intent, described in natural language (Mueller et al., 2024; Sun et al., 2024; Li et al., 2023a; Jain et al., 2024; Wang et al., 2024a). A similar effect occurs when Code-LMs are prompted to generate code in multiple phases with the aid of code sketches that break down complex compositional syntactic structures (Guo et al., 2022b; Zan et al., 2022). Attempts at similarly untwining the two channels at pre-training time using contrastive learning (Wang et al., 2022) or translation (Chakraborty et al., 2022) between semantically equivalent code fragments have been hamstrung by the limited attainability and diversity of such data (Bui et al., 2021). Other attempts have resorted to architectural choices that are hard to scale (Zhang et al., 2021a) or cannot be utilized when text and code are intertwined (Kim et al., 2021), which is the most common mode of operation for Code-LMs. Thus, there is a need for an objective that fosters both semantic and syntactic code understanding while being easy to incorporate into modern autoregressive pre-training of Code-LMs.

**Surmounting the code *data-wall*.** The choice of model and corpus size in language model pre-training has traditionally been guided by *scaling laws* (Kaplan et al., 2020; Hoffmann et al., 2022) which provide a recipe on how to best trade the two off under a fixed computational budget. However, the widespread adoption of LMs has warranted a revision of scaling laws to account for inference costs and latency and favours smaller models trained on more data (Sardana et al., 2024). With quality filtering in place, transformer-based models have demonstrated the ability to keep learning on orders of magnitude more data than what was considered optimal according to scaling laws (Gadre et al., 2024). This finding has subsequently been validated for Code-LMs in both language modelling (Rozière et al., 2023) and downstream task performance (Liu et al., 2024; Li et al., 2023b; Allal et al., 2023).

Consequently, scaling up high-quality corpora for LM pre-training has hit the *data-wall* [1] — the exhaustion of openly accessible, unique and high-quality data (Bi et al., 2024). Recent LM releases (Dubey et al., 2024) may have already hit the limit of clean and openly crawled data (Penedo et al., 2024) ($\approx$15 trillion tokens). Code-LM developers face an even more alarming data-constrained reality, with the most extensive corpus of freely licensed source code containing under a trillion tokens by most measures (Lozhkov et al., 2024). Attempts to circumvent this problem include repeating data (Muennighoff et al., 2023), synthesizing data (Gunasekar et al., 2023; Wei et al., 2024) and obtaining code-adjacent data from natural language corpora using domain-classifier verdicts (Liu et al., 2024). Further attempts have sought to obtain more data through the interplay between model-generated code and programming language toolchains (Ding et al., 2024; Zheng et al., 2024). Nonetheless, the best means to source the next trillion tokens of code data remains an open question.

**The case for (de-)obfuscation as an objective.** We posit that using obfuscated code in Code-LM pre-training enables: **1)** Crafting objectives that better disentangle syntactic and semantic understanding of code; **2)** A way out of the code data bottleneck. Obfuscation refers to syntactically mangling source code files in a manner that preserves their semantic correctness, primarily as a measure to prevent adversaries from reverse-engineering its function (Schrittwieser et al., 2016). This is mainly achieved by substituting variable, function, class and namespace identifiers with uninformative, generic names that make it harder for readers to discern semantic intent (Lawrie et al., 2006). We provide complete examples of obfuscated code in Appendix B.3.

Specifically, we introduce a translation objective for pairs of the original source code and corresponding obfuscated code for a part of the pre-training corpus, including training the Code-LM explicitly on the structure of the obfuscated code itself, in contrast to prior work that leverages de-obfuscation objectives (Lachaux et al., 2021). Stripping away these natural language components in code (e.g., variable and class names) that reveal semantic intent (Petrescu et al., 2023) forces the Code-LM to reason about the algorithmic meaning of the code from the syntactic structure alone, without allowing it to exploit semantic shortcuts. Simultaneously, the translation objective grounds the Code-LM's

---

[1] 🌐 situational-awareness.ai/from-gpt-4-to-agi/#The_data_wall

understanding of the obfuscated code in the original source code, ensuring no modifications to the Code-LM are needed for inference. Additionally, obfuscation can be performed stochastically, thus providing an easy way to scale up code pre-training corpora with diverse semantic-preserving transformations in a way existing rule-based techniques cannot match.

**Contributions and research questions.** In this work, we: **1)** create `ObscuraX`, a source-to-obfuscated-code translation pairs dataset containing approximately 55M pairs in seven programming languages; **2)** conduct a systematic investigation of semantically grounding Code-LMs in obfuscated code, demonstrating sizeable and consistent empirical gains across a broad range of tasks and programming languages; **3)** train `ObscuraCoder`, a suite of base Code-LMs pre-trained on a data mix containing `ObscuraX`, ranging from in size 255M to 2.8B parameters.

Our study of obfuscation-based semantic grounding for training Code-LMs is structured as follows:

- **RQ1:** Does training on obfuscated code help improve the performance of Code-LMs on tasks requiring (a) syntactic and (b) semantic code understanding?
- **RQ2:** Can supplementing identifier obfuscation with import obfuscation improve Code-LMs on library-oriented [2] code generation?
- **RQ3:** Are there any (undesirable) side-effects of obfuscation training, i.e. can it lead to performance regression on tasks on which causal LM excel, e.g., code-completion?
- **RQ4:** How does the relative performance of obfuscation-grounded training differ and scale with increasing model size?

## 2   RELATED WORK

We briefly outline four relevant lines of existing work: **1)** syntactically- and semantically-aware code representation learning; **2)** code translation as a pre-training objective; **3)** code representation learning based on programming language toolchains; and, most related, **4)** code (de-)obfuscation.

**Syntactically- and semantically-aware code representation learning.** The conventional autoregressive objective used to train Code-LMs mimics their natural language counterparts, offering the convenience of reusing the same data preparation and training pipelines. This, however, fails to capture all the nuances of programming language syntax (Velasco et al., 2024), with resulting Code-LMs displaying uncertainty over certain syntactic structures such as exception handling and type inference (Palacio et al., 2023). Prior bids to address this issue at training time have resorted to ensuring syntactically faithful generations from Code-LMs using external signals via reinforcement learning (Parsert & Polgreen, 2024) or by training dedicated syntax verification models (Yang et al., 2023b), *inter alia*. Other approaches include (i) targeting input embeddings by deploying differentiable adversarial perturbations for improved robustness (Li et al., 2022a), (ii) incorporating auxiliary syntax tree position (Guo et al., 2022c) and node-type information (Park et al., 2023; Takerngsaksiri et al., 2024; Saberi & Fard, 2023) and (iii) test-time interventions such as syntactic grammar constrained decoding (Yang et al., 2023a; Shen et al., 2024).

Similarly, while causal LM-ing imparts some understanding of code semantics (Ahmed et al., 2023b), code representations learnt this way often fail on tests of semantic equivalence for complex code fragments (Troshin & Chirkova, 2022). Existing work has thus sought to explicitly encode the semantics of the source code by leveraging data flow graphs (Chae et al., 2017), control flow graphs (David et al., 2020), program dependence graphs (Bichsel et al., 2016) and compiler intermediate representations (Ben-Nun et al., 2018). Attempts along these lines usually employ task-specific graph-based (Du & Yu, 2023; Liu et al., 2023c; Pei et al., 2024; Guo et al., 2021), tree-based (Jiang et al., 2022b), tensor-based (Yang et al., 2023c) or hybrid (Shi et al., 2023; Wang, 2019) architectures and objectives. Other work seeks to improve semantic understanding by training Code-LMs on runtime execution traces (Wang & Su, 2019; Wang et al., 2023a; Liu et al., 2023a). In contrast, we show that teaching a model the code structure via obfuscated code while simultaneously training it to deobfuscate that code improves Code-LMs' syntactic and semantic code understanding abilities.

**Translation as a pre-training objective.** Most mainstream Code-LMs (Lozhkov et al., 2024; Liu et al., 2024; Mishra et al., 2024; Zhao et al., 2024)—by virtue of being trained on GitHub code—are at least implicitly grounded in comment text- to-code translation objectives. However, they have

---

[2]In this work, library-oriented refers to non-object-oriented Code Function Call APIs, as per the `Gorilla OpenFunctions` taxonomy. Typical examples are external packages like Numpy, NetworkX, Seaborn, etc.

a mixed record on code-to-code translation (Pan et al., 2024). Attempts to improve code-to-code translation have trained Code-LMs on parallel data, usually sourced from programming contests (Zhu et al., 2022b;a). Subsequent work tried to generalize parallel data collection implicitly via, e.g., back-translation (Ahmad et al., 2023; Chen & Lampouras, 2023; Rozière et al., 2020) with applications in code repair (Drain et al., 2021; Silva et al., 2023) and code explanation (Mahbub et al., 2023).

Translation objectives have also been employed in training to improve Code-LMs' application-specific representations. These include semantic grounding in (i) synthetic code outlines for multi-step generation (Shi et al., 2024), (ii) syntax tree leaf nodes for retrieval (Phan & Jannesari, 2023), (iii) compiled binary code for vulnerability detection (Ahmad & Luo, 2024), (iv) pseudo-code for code comprehension (Oda et al., 2015) and (v) compiler intermediate representations for improved multilingual performance (Szafraniec et al., 2023). We extend this body of work by demonstrating the benefits of obfuscation-based translation.

**Programming language toolchain grounded representation learning.** There is an abundance of work that grounds code in artefacts that originate from various stages of compilation. Amongst the frontend artefacts, attempts to leverage syntax are the most common, most often encoding the (linearization of) syntactic trees with recurrent (Jiang et al., 2022a), convolutional (Mou et al., 2016), graph-based (Zhang et al., 2022) and transformer-based models (Guo et al., 2022a). Other approaches (i) use syntactic trees as a search prior for graph-based decoding (Brockschmidt et al., 2019). (ii) use syntactic tree leaf element boundaries to inform masking (Wang et al., 2021b) and tokenization (Gong et al., 2024), (iii) predict (heuristically selected) paths from the tree as an auxiliary pre-training objective (Tipirneni et al., 2024) and, (iv) leverage syntactic trees for data augmentation: heuristic generation of semantic-preserving transformations, leveraged for contrastive learning (Jain et al., 2021; Wang et al., 2021a; Bahrami et al., 2021). Further compiler frontend artefacts such as data flow graphs (Brauckmann et al., 2020) and control flow graphs (Nair et al., 2020) have also been leveraged to ground program understanding. Other work (Shojaee et al., 2023; Le et al., 2022) uses comparisons of data flow and control flow graphs between the generated and reference code to shape a reward function for reinforcement learning-guided generation.

Compiler intermediate representations have also been leveraged for grounding, as (i) a source of meaning-preserving transformations (Li et al., 2022c) and as (ii) a means to improve multilingual performance of Code-LMs (Paul et al., 2024). Further from the developer, compiler backend artefacts such as diagnostics (Ahmed et al., 2023a) and textual feedback (Ren et al., 2024; Liu et al., 2023b) have been utilized to reduce functionally erroneous Code-LMs outputs. In this work, we leverage obfuscated code, typically produced by the compilers' backend, for improved grounding and dual-channel (i.e., syntax vs. semantics) disentanglement. However, aiming for customizability and broad applicability across programming languages, we write our own custom obfuscator in this work.

**Code (de-)obfuscation.** Code obfuscation is the semantic-preserving modification of source code to hide its execution intent. Neural approaches mainly focus on control flow (Ma et al., 2014), data (Yala et al., 2022) and identifier (Zhou et al., 2022; Datta, 2021) obfuscation but have limited adoption due to their tendency to introduce performance regressions and execution faults (Skolka et al., 2019).

In contrast, due to its underspecified and formally intractable nature (Takang et al., 1996), code de-obfuscation is a natural task for neural methods. In particular, probabilistic (Raychev et al., 2019; Alon et al., 2018), recurrent (Bavishi et al., 2018; Lacomis et al., 2019) and transformer-based (David et al., 2020) models have all been employed for code de-obfuscation. More recently, DOBF (Lachaux et al., 2021) pioneered de-obfuscation as a pre-training objective for encoder-decoder Code-LMs, rendering it more effective than traditional sequence-to-sequence denoising objectives (Raffel et al., 2020) on many-shot translation and retrieval tasks. DOBF, however, only trains the Code-LM on the identifier map and does not backpropagate gradients w.r.t. the source or the obfuscated code; instead, it relies on initialization with a model trained on other objectives to maximize its code understanding abilities. In contrast, our `ObscuraCoder` models are trained from scratch on a bidirectional translation objective while mixing in language modelling on the unpaired source and obfuscated code samples. To the best of our knowledge, we are the first to show that (de-)obfuscation translation is a viable pre-training objective for improving modern multilingual decoder-only Code-LMs. Explicitly modelling the structure of the original and obfuscated code, `ObscuraCoder` yields significant improvements in semantic robustness and zero-shot code completion compared to both vanilla autoregressive LMs and (a decoder-only variant of) DOBF. Moreover, we are the first to show that trading off semantic correctness (for a fraction of the obfuscated samples) by mangling import statements in addition to identifiers leads to large gains in library-oriented code generation.

# 3 OBSCURAX: SOURCE TO OBFUSCATED CODE TRANSLATION PAIRS

Figure 1: An overview of the `ObscuraX` data sourcing (Step 1 and 2) and the `ObscuraCoder` pre-training objective, contrasted against that of causal LM and DOBF (Step3). Notably, `Obscu-raCoder` does not mask the obfuscated tokens and additionally trains on samples comprising only source or obfuscated data.

Seeking to test the hypotheses we posit in Section 1, we construct a multilingual source-code-to-obfuscated-code parallel dataset. We initiate this effort by acquiring source code files containing fewer than 2000 lines of code from the Stack corpus (Kocetkov et al., 2023) in seven languages — C, C++, Go, Java, Python, Rust and TypeScript. We ensure to retain all code sourced from research repositories [3] and accepted programming contest solutions (Puri et al., 2021), owing to their tendency to be self-contained. The resultant corpus is MinHash (Broder, 1997) de-duplicated with the surviving source files' syntax trees extracted using Tree-sitter [4]. Well-known for its fault-tolerant approach to parsing, Tree-sitter affords us a reliable way to handle open-domain source code of vague provenance whose executability and correctness are not assured.

The syntax trees obtained contain the span indices of the code's constituent syntactic elements. However, the targeted obfuscation of specific categories of syntactic structures requires reliably mapping this category information to the sub-trees of the syntax tree. Hence, we write a customized obfuscator that leverages Tree-sitter's own tree query language [5]. We build on top of queries employed by popular IDEs [6] to account for the inherent complexity in accurately tracking syntactic structure compositions and their scopes for the multi-paradigm programming languages in `ObscuraX`.

The consequent mapping between spans in the source files and syntactic element category is employed to create a mapping from the original variable, function and class names to their obfuscated counterparts. This is stochastic and is influenced by the obfuscation proportion hyper-parameter $p_{obf}$, which informs the proportion of the original elements we wish to obfuscate. In practice, $p_{obf}$ is uniformly varied up to 0.9, with higher values avoided to provide the model with some information to learn the deobfuscation in a principled manner. We limit obfuscation to 150 members of a specific syntactic category to reasonably restrict the samples' de-obfuscation difficulty level. The processed

---

[3] AlgorithmicResearchGroup/arxiv_research_code
[4] git tree-sitter/tree-sitter
[5] git tree-sitter-grammars/tree-sitter-query
[6] git nvim-treesitter/nvim-treesitter

counterparts are formatted as `VAR_{n}`, `FUNC_{n}` and `CLASS_{n}` respectively, where n ranges from 0 to 149. Finally, for cases where the same name is shared by different syntactic structures or by syntactic structures of the same variety but different scope (e.g. global and local variables), we preserve their shared semantic grounding using a catch-all category formatted as `ID_{n}`.

The described process is fully correctness-preserving, allowing the curation of learning objectives that better disentangle the syntactic and semantic channels. However, to improve library-oriented code generation, we sacrifice this property for ≈25% of the samples and obfuscate the imports as `IMPORT_{n}`. This, in turn, facilitates the Code-LM's understanding of API usage during de-obfuscation, which can be vital for its ability on infrequently invoked libraries or rare use-cases (Rabin et al., 2023). Our resultant dataset, which we dub `ObscuraX`, comprises ≈55M samples and is the largest multilingual collection of source code to obfuscated code translation pairs yet. Figure 1 details a high-level view of how `ObscuraX` is a critical part of the `ObscuraCoder` pre-training pipeline and also lists some examples. Refer to Appendix B.3 for more detailed samples in all languages.

## 4 EXPERIMENTAL SETUP

**Pre-training data recipe.** We compile two comparable pre-training corpora to facilitate a fair comparison between `ObscuraCoder` and standard autoregressive pre-training. We begin by obtaining 105B tokens of high-quality filtered and de-duplicated text data, with the sourcing biased towards informative and instructional content from sources such as books, wikis, news articles and research papers following existing literature (Penedo et al., 2024). This is further augmented with 15B tokens of code-adjacent text and code-text data such as library documentation, coding tutorials, GitHub commits and issues. We dub this 120B-token corpus **filtered-code-text**. We then collect source code data from the Stack V1 (Kocetkov et al., 2023) corpus. Specifically, we select only the splits of languages from `ObscuraX`, i.e., C, C++, Go, Java, Python, Rust and TypeScript. We also select the Markdown split because of its importance for Code-LMs' instruction-following abilities (Luo et al., 2024). We subject this code corpus to further quality filtering and de-duplication, obtaining the final 76B-token corpus we term **filtered-source-code**.

We organize the causal LM pre-training corpus into two phases: (1) 90B tokens sampled from `filtered-code-text`, and (2) two copies of `filtered-source code` (152B tokens) plus the remaining 30B tokens of `filtered-code-text` randomly shuffled; for a total training data of ≈272B tokens. By contrast, the `ObscuraCoder` pre-training corpus reuses the first phase of causal LM training but compiles the second phase as a random shuffle of (1) 64B tokens from `filtered-source code`, (2) 30B tokens of obfuscated code from `ObscuraX`, (3) 58B tokens of translation pairs from `ObscuraX` and (4) the (same) remaining 30B tokens of `filtered-code-text`; also totalling ≈272B tokens. The translation pairs from `ObscuraX` are separated by two sentinel tokens which we add to the Code-LM's vocabulary—`<src_to_obf>` and `<obf_to_src>`—used for the respective translation directions (see Figure 1), which are each sampled for 50% of the instances.

Both the corpora are designed to follow existing Code-LM best practices of initially priming the model on text-only data before training on a code-text mixture dominated by source code data (Guo et al., 2024; Rozière et al., 2023; Hernandez et al., 2021; Tao et al., 2024). Similarly, when shuffling the various splits during corpora creation, we ensure no source code fragment is included more than three times, thus staying within the optimal learning regime (Muennighoff et al., 2023). Finally, we ensure that our corpora are n-gram decontaminated w.r.t. downstream tasks on which we evaluate (Chen et al., 2021). Appendix B.2 further details our data sourcing and filtering pipeline.

**Pre-training details.** For direct comparability between vanilla Code-LM pre-training and `ObscuraCoder`, we train both using the Llama architecture (Rozière et al., 2023) in four model sizes: 255M, 491M, 1.2B and 2.8B parameters. While most frontier models are pre-trained on corpora roughly an order of magnitude larger than ours, we argue that the scale of our pre-training runs suffice to demonstrate the merits of obfuscation-based semantic grounding, as they are nearly an order of magnitude greater than what would be deemed optimal by the scaling-laws (Hoffmann et al., 2022).

All training runs are performed using an open-source implementation [7] of the Megatron-LM (Shoeybi et al., 2019) kernels and leverage Deepspeed Stage-2 (Rajbhandari et al., 2020) sharding on BF16 precision. We use the Adam optimizer (Kingma & Ba, 2015), with a learning rate of 5e-4 and a cosine annealed schedule that terminates at 5% of the peak learning rate. The models are trained

---

[7] `git EleutherAI/gpt-neox`

using FlashAttention-2 (Dao, 2024) with a sequence length of 2048 tokens and a batch size of 256 for 520K steps. We use a custom byte-level BPE tokenizer (Wang et al., 2020) with a vocabulary size of 49152 tokens in total that we train on a 5B token corpus comprising of code and code-adjacent text data. The tokenizer is augmented with special tokens pertaining to the outputs of our obfuscator — `ID_{n}`, `CLASS_{n}`, `FUNC_{n}`, `VAR_{n}` and `IMPORT_{n}` — n ranging from 0 to 149.

**Inference details.** For inference, we resort to a Paged Attention (Kwon et al., 2023) enabled fork of an open-source evaluation harness.[8] We conduct our evaluation using model checkpoints loaded in half-precision (for efficiency) with nucleus sampling ($p$=0.9). All inference runs are conducted on Nvidia A100 80GB GPUs with 95% of the GPU VRAM explicitly reserved for vLLMs GPU pages. We further set aside 64GB of RAM as a CPU swap, allowing for offloading pages to the CPU during bursts of long sequences. We additionally limit the continuous batching parameter to 32.

## 5 EXPERIMENTAL RESEARCH QUESTIONS & RESULTS

Our evaluation comprises a mix of five zero-shot and fine-tuning tasks, selected to provide answers to the three research questions from Section 1. For each task, we compare the performance of `ObscuraCoder` against an equally-sized autoregressive LM. We further contextualize the sample-efficiency benefits of our obfuscation objective by including comparisons to seven frontier Code-LMs from the Deepseek-Coder (Guo et al., 2024), CodeGemma (Zhao et al., 2024), Phi (Gunasekar et al., 2023) and StarCoder (Lozhkov et al., 2024; Li et al., 2023b) families that are under 3B parameters in size and pre-trained on corpora between 5x to 22x larger than the one used to train `ObscuraCoder`.

For fine-tuning tasks, all models are trained for three epochs using a cosine scheduler with a peak learning rate of 5e-5 using LoRA (Xu et al., 2024) modules coupled with trainable embeddings. We follow prior work (Poth et al., 2023) and train LoRA modules with rank 64 for classification tasks and 256 for open-ended generation. Unless otherwise specified, we use greedy decoding at inference.

| ⌨Model | Pre-Training | Defect | | ReCode pass@1 | | | BigCodeBench | | |
|---|---|---|---|---|---|---|---|---|---|
| | Token Count | Acc. | F1-Score | Format | Syntax | Function | Pass@1 | Pass@10 | Pass@25 |
| CausalLM 255M | 272B | 62.39 | 61.11 | 11.37 | 8.98 | 3.13 | 2.12 | 3.87 | 4.32 |
| ObscuraCoder 255M | 272B | 62.80 | 62.14 | 14.17 | 11.80 | 4.95 | 2.64 | 4.45 | 4.97 |
| | | +0.41 | +1.03 | +2.80 | +2.82 | +1.82 | +0.52 | +0.58 | +0.65 |
| CausalLM 491M | 272B | 63.36 | 62.88 | 15.04 | 11.30 | 5.22 | 2.83 | 6.29 | 8.91 |
| ObscuraCoder 491M | 272B | 64.27 | 63.72 | 23.88 | 20.04 | 8.01 | 3.02 | 7.09 | 11.97 |
| | | +0.91 | +0.84 | +8.84 | +8.74 | +2.78 | +0.19 | +0.80 | +3.06 |
| CausalLM 1.2B | 272B | 63.49 | 63.59 | 27.65 | 21.01 | 11.30 | 7.98 | 14.18 | 23.77 |
| ObscuraCoder 1.2B | 272B | 64.49 | 64.76 | 36.37 | 27.86 | 16.66 | 9.67 | 17.89 | 28.04 |
| | | +1.00 | +1.17 | +8.73 | +6.85 | +5.36 | +1.69 | +3.71 | +4.27 |
| CausalLM 2.8B | 272B | 64.73 | 64.36 | 34.70 | 26.20 | 14.72 | 10.79 | 22.87 | 31.95 |
| ObscuraCoder 2.8B | 272B | 65.58 | 65.42 | 45.29 | 35.97 | 20.11 | 14.77 | 31.94 | 48.78 |
| | | +0.85 | +1.06 | +10.59 | +9.77 | +5.38 | +3.98 | +9.07 | +16.83 |
| StarCoderBase 1B | 1.0T | 64.96 | 64.68 | 28.97 | 25.92 | 13.84 | 5.83 | 13.58 | 22.77 |
| DeepSeekCoder 1.3B | 2.0T | 65.21 | 64.59 | 48.04 | 44.42 | 25.27 | 19.57 | 34.18 | 47.84 |
| StarCoderBase 3B | 1.0T | 65.82 | 64.92 | 37.21 | 30.25 | 17.49 | 8.26 | 26.79 | 37.74 |
| StarCoder2 3B | 3.3T | 66.37 | 65.55 | 53.82 | 47.66 | 24.58 | 16.38 | 36.11 | 48.81 |
| StableCode 3B | 5.6T | 62.31 | 61.71 | 50.09 | 43.60 | 24.15 | 12.98 | 31.75 | 49.44 |
| CodeGemma 2B | 3.5T | 61.93 | 60.57 | 10.31 | 9.62 | 7.04 | 17.39 | 33.12 | 43.66 |
| Phi-2 | 1.4T | 64.56 | 63.97 | 59.99 | 53.71 | 32.71 | 21.83 | 38.54 | 53.78 |

Table 1: Results table for **RQ1** and **RQ2** comparing syntactic understanding for code defect detection on CodeXGLUE (Lu et al., 2021), semantic understanding for robust code completion on ReCode (Wang et al., 2023b) and library-oriented code generation on BigCodeBench (Zhuo et al., 2024). For detailed split-level breakdowns in ReCode results refer to Tables 7 to 9 in Appendix C.

**RQ1: Obfuscation translation → better syntactic and semantic understanding?**

We first test if our obfuscation-based training leads to a better understanding of complex semantic and syntactic structures, which are crucial for solving complex tasks like detecting vulnerable code.

**CodeXGLUE Code defect detection.** Detecting code defects requires integrating information from both channels due to the inherent dual-channel nature of software bugs. While some bugs, such as insecure argument usage, demand a grasp of syntactic structures (Yamaguchi et al., 2014; Ray et al., 2016) and are easier to detect when code syntax is simplified (Thummalapenta & Xie, 2011), others, like memory leaks and null pointer references, require an understanding of control and data flow semantics (Zhou et al., 2019; Wang et al., 2016). We benchmark `ObscuraCoder` and baseline CodeLMs on a binary defect detection classification task from CodeXGLUE (Lu et al., 2021).

---

[8] git `bigcode-project/bigcode-evaluation-harness`

**ReCode.** We next employ five differently seeded ReCode (Wang et al., 2023b) transformations of HumanEval (Chen et al., 2021). These examine a Code-LM's (zero-shot) robustness to semantically-preserving syntactic perturbations based on its greedy decoding completions on code generation prompts mangled using an extensive battery of Format, Function, and Syntax transforms, thus explicitly testing the model's understanding of the interplay between code syntax and semantics.

**Results.** Table 1 summarizes the results of the above two tasks. We observe that obfuscation-based objectives of ObscuraCoder consistently bring significant performance gains on top of causal language modelling for all model sizes and both tasks. The gains are particularly large on the ReCode task, keeping with prior work, which reports that translation between semantically equivalent programs benefits understanding of code intent (Guizzo et al., 2024). The ObscuraCoder models frequently match or surpass the performance of the next-size CausalLMs, and ObscuraCoder 2.8B beats several frontier models on semantic code understanding.

### RQ2: Does code obfuscation training lead to better library-oriented code generation?

**BigCodeBench.** We investigate the effects of our de-obfuscation objective taken to its logical extreme, including the choice to obfuscate imports and package names, on Code-LMs' abilities in library-oriented code generation. Such obfuscation, we hypothesized, should have forced Code-LMs to understand what library APIs do and when to invoke them. Prior work (Zhang et al., 2021b) points to the demanding nature of package de-obfuscation. Because of this, we evaluate our Code-LMs zero-shot on the BigCodeBench (Zhuo et al., 2024) benchmark [9] in the completion setting, thus testing models' competence in API usage. We sample 50 generations per problem and report the pass@k performance for $k \in \{1, 10, 25\}$. The pass@1 performance emulates usage scenarios where correctness is paramount: we thus decode with temperature sampling with a low temperature of 0.1. In contrast, pass@10 and pass@25 estimates correspond to scenarios where creativity and diversity of generations are more important: we thus use a higher sampling temperature of 0.8.

**Results.** The results in the BigCodeBench column of Table 1 demonstrate substantial gains in library-oriented code generation performance facilitated by obfuscation-grounded pre-training that includes obfuscation of imports and package names. We believe that our design choice to represent obfuscated packages, which are often multi-token strings, with a single (special) token is one key driver of these gains, in line with suggestions from prior work (Hadi et al., 2022). Particularly encouraging is the observation that ObscuraCoder's library-oriented generation gains (over the vanilla autoregressive LM-ing) widen with increasing model size. ObscuraCoder represents a significant advancement in addressing API misuse, a major obstacle to wider Code-LM adoption (Wang et al., 2024b). Promisingly, ObscuraCoder 2.8B is competitive with the best frontier open-source models in this challenging setting that most closely mimics real-world usage.

### RQ3: Are there negative side-effects of obfuscation-based pre-training?

We next test whether improvements in dual-channel understanding and library-oriented code generation come at a cost, i.e., performance deterioration, for zero-shot multilingual code completion, which is still the most common application of Code-LMs.

**Multipl-E.** We first carry out multlingual evaluation of pass@k performance for $k \in \{1, 10, 100\}$ in five languages (C++, Java, Python, Rust and TypeScript) using the Multipl-E benchmark (Cassano et al., 2023). We sample 200 generations per problem. Like in BigCodeBench evaluation, we set the sampling temperature to 0.1 for pass@1 and to 0.8 for pass@10 and pass@100.

**CommitChronicle.** As a further measure of multilingual code competence, we evaluate Code-LMs on CommitChronicle (Eliseeva et al., 2023), a fine-tuning code-change summarization benchmark in seven languages (C, C++, Go, Java, Python, Rust and TypeScript). Summarizing code changes is a good proxy for multilingual code understanding due to its dual nature w.r.t. code completion (Wei et al., 2019) and the tendency of the capabilities in each task—text-to-code and code-to-text—to reinforce the other. We construct language-specific splits by filtering the original dataset and partitioning 75%, 15% and 10% of the data into train, validation and test splits, respectively. In fine-tuning, we ensure that autoregressive losses are backpropagated only for the target summaries.

**Results.** Table 2 points to the strong performance of ObscuraCoder on both multilingual benchmarks. Not only do our (de-)obfuscation pre-training objectives not hurt multilingual generation

---

[9]As early adopters, we ran v0.1.0 of the benchmark, where 13 network usage problems were found to contain unreliable tests and were discarded. We report all results on the remaining 1127 problems.

| ♻ Model | Pre-Training | | Multipl-E | | | Commit Chronicle | | |
|---|---|---|---|---|---|---|---|---|
| | Token Count | | Pass@1 | Pass@10 | Pass@100 | ROUGE-1 | ROUGE-2 | ROUGE-L |
| CausalLM 255M | 272B | | 4.29 | 7.13 | 14.36 | 32.76 | 10.15 | 31.14 |
| ObscuraCoder 255M | 272B | | 5.93 | 9.66 | 18.20 | 33.40 | 10.64 | 31.70 |
| | | | +1.65 | +2.53 | +3.84 | +0.65 | +0.49 | +0.56 |
| CausalLM 491M | 272B | | 5.84 | 10.64 | 20.69 | 34.08 | 11.02 | 32.40 |
| ObscuraCoder 491M | 272B | | 8.76 | 14.33 | 25.86 | 34.85 | 11.44 | 33.03 |
| | | | +2.92 | +3.69 | +5.17 | +0.78 | +0.42 | +0.63 |
| CausalLM 1.2B | 272B | | 12.07 | 19.50 | 34.10 | 35.65 | 11.96 | 33.79 |
| ObscuraCoder 1.2B | 272B | | 18.34 | 29.34 | 47.79 | 37.00 | 13.20 | 35.34 |
| | | | +6.27 | +9.83 | +13.69 | +1.35 | +1.24 | +1.55 |
| CausalLM 2.8B | 272B | | 16.70 | 28.83 | 47.53 | 36.72 | 12.89 | 35.07 |
| ObscuraCoder 2.8B | 272B | | 23.55 | 39.41 | 61.93 | 38.07 | 14.15 | 36.60 |
| | | | +6.85 | +10.59 | +14.39 | +1.35 | +1.26 | +1.53 |
| StarCoderBase 1B | 1.0T | | 13.12 | 22.63 | 37.78 | 37.05 | 13.19 | 35.30 |
| DeepSeekCoder 1.3B | 2.0T | | 26.81 | 47.46 | 71.70 | 35.86 | 12.95 | 34.42 |
| StarCoderBase 3B | 1.0T | | 19.21 | 33.48 | 57.60 | 38.65 | 14.44 | 36.88 |
| StarCoder2 3B | 3.3T | | 27.38 | 49.63 | 72.86 | 38.75 | 14.58 | 36.96 |
| StableCode 3B | 5.6T | | 26.73 | 47.93 | 68.88 | 38.08 | 14.35 | 36.69 |
| CodeGemma 2B | 3.5T | | 22.71 | 37.98 | 62.14 | 36.14 | 13.29 | 35.28 |
| Phi-2 | 1.4T | | 22.27 | 39.96 | 57.30 | 35.24 | 12.16 | 33.61 |

Table 2: Results table for **RQ3** comparing multilingual code completion averages on Multipl-E (Cassano et al., 2023) (C++, Java, Python, Rust, TypeScript) and multilingual commit summarization averages on CommitChronicle (Eliseeva et al., 2023) (C, C++, Go, Java, Python, Rust, TypeScript). For detailed language-wise breakdowns in Multipl-E results refer to Tables 10 to 12 and CommitChronicle results refer to Tables 13 to 15 in Appendix C.

performance compared to causal LM-ing, but they seem to improve it. Moreover, similar to the results on BigCodeBench, the gains from obfuscation pre-training over autoregressive LM-ing generally increase with the model size. Prior work (Bavarian et al., 2022; Wang et al., 2021b) refers to incorporation of additional objectives during pre-training as being "for free" when these objectives do not incur any deterioration in autoregressive performance. Owing to its pre-training objective, `Obscura­Coder` goes beyond this and actually displays strong evidence of positive transfer from improved syntactic, semantic and API understanding to multilingual code completion and summarization.

**RQ4: Do the benefits of obfuscation-grounding scale?**

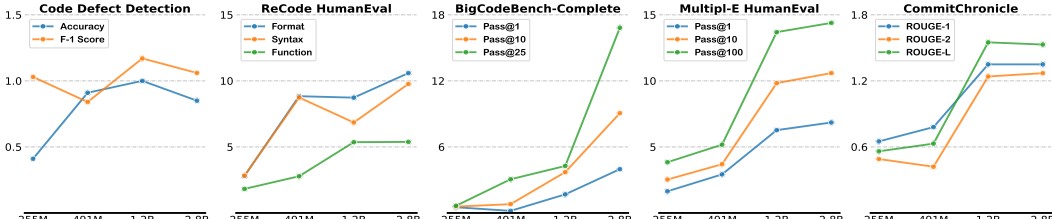

Figure 2: Downstream performance gains of `ObscuraCoder` viz-a-viz Causal Code-LMs.

Due to computational constraints, we only train models up to 2.8B parameters in size. We attempt to extrapolate the effects of obfuscation pre-training to parameter sizes resembling those of the most widely adopted Code-LMs [10] by investigating the scaling trends of downstream task performance.

Figure 2 summarizes the performance of `ObscuraCoder` across all five downstream tasks as a function of model size. Expectedly, scaling model size brings more modest gains in fine-tuning setups (CodeXGLUE Defect Detection and CommitChronicle) than in zero-shot evaluations: we observe saturation in fine-tuning performance also for the larger frontier model (see Table 1 and Table 2). Scaling trends are very prominent in zero-shot tasks, especially library-oriented (BigCodeBench) and multilingual (Multipl-E) generation: this is encouraging, as these tasks correspond to Code-LMs' most prevalent modes of use. Finally, we observe the strongest scaling trends in settings that reward diverse code generation: high pass rate estimates on high-temperature generations. This is promising for generate-then-rerank settings, where being able to choose from a deep bank of valid solutions drives further improvements to end-user utility (Li et al., 2022b; Chen et al., 2023) and opens room for optimizing non-functional axes (Waghjale et al., 2024).

Generally, our results strongly agree with prior work that establishes the benefits of information-preserving transforms in LM pre-training (Yuan & Liu, 2022; Maini et al., 2024). Specifically, we establish that Code-LMs can substantially benefit from (de-)obfuscation-based translation objectives.

---

[10] At the time of writing, this ranges from approx. 3x to 150x larger than the largest `ObscuraCoder` model.

## 6 ABLATIONS

**How Does ObscuraCoder Compare Against DOBF?** We attempt an explicit head-to-head comparison between the de-obfuscation objective pioneered by DOBF (Lachaux et al., 2021) and our obfuscation-grounded translation objective. However, the originally released DOBF model is an encoder-decoder model with 126M parameters, making it half the size of our smallest model. Hence, to facilitate a fair comparison, we train a decoder-only adaptation of DOBF on a modified version of the ObscuraCoder corpus of ≈272B tokens outlined in Section 4. Specifically, the 30B tokens of obfuscated code and the 58B tokens of translation pairs are replaced by 88B tokens of source code to identifier map translation pairs sourced from ObscuraX (see Figure 1 for the DOBF objective).

We train size-matched versions of DOBF on the above-detailed corpus. Staying faithful to the original de-obfuscation objective, we mask the obfuscated code from the loss computation and only back-propagate gradients for the identifier maps targets.[11] [12] Results in Table 3 show that ObscuraCoder has an advantage over DOBF across the board, with particularly prominent gaps on on zero-shot tasks. Promisingly, the performance gap appears to widen with increasing model size.

| ⏱ Model | Defect | ReCode | | | BigCodeBench | | Multipl-E | | CC |
|---|---|---|---|---|---|---|---|---|---|
| | F-1 Score | Format | Syntax | Function | Pass@1 | Pass@25 | Pass@1 | Pass@100 | ROUGE-2 |
| ObscuraCoder 255M | 62.14 | 14.17 | 11.80 | 4.95 | 2.64 | 4.97 | 5.93 | 18.20 | 10.64 |
| DOBF 255M | 62.22 | 14.00 | 11.67 | 4.97 | 2.68 | 4.79 | 4.62 | 16.85 | 10.36 |
| | +0.08 | -0.17 | -0.13 | +0.02 | +0.04 | -0.18 | -1.31 | -1.35 | -0.28 |
| CausalLM-CP 255M | 62.03 | 11.05 | 8.27 | 2.81 | 1.86 | 4.21 | 4.00 | 15.10 | 10.08 |
| | -0.11 | -3.12 | -3.53 | -2.14 | -0.78 | -0.76 | -1.93 | -3.10 | -0.56 |
| ObscuraCoder 491M | 63.72 | 23.88 | 20.04 | 8.01 | 3.02 | 11.97 | 8.76 | 25.86 | 11.44 |
| DOBF 491M | 63.65 | 15.93 | 12.70 | 6.91 | 2.89 | 10.06 | 6.76 | 20.32 | 11.15 |
| | -0.07 | -7.95 | -7.33 | -1.10 | -0.13 | -1.91 | -2.00 | -5.54 | -0.29 |
| CausalLM-CP 491M | 62.27 | 15.59 | 11.91 | 6.18 | 2.46 | 8.12 | 6.89 | 22.04 | 10.92 |
| | -1.45 | -8.29 | -8.13 | -1.83 | -0.56 | -3.85 | -1.87 | -3.82 | -0.52 |
| ObscuraCoder 1.2B | 64.76 | 36.37 | 27.86 | 16.66 | 9.67 | 28.04 | 18.34 | 47.79 | 13.20 |
| DOBF 1.2B | 64.67 | 30.06 | 23.65 | 13.07 | 9.14 | 25.95 | 15.25 | 40.30 | 12.52 |
| | -0.09 | -6.31 | -4.21 | -3.59 | -0.53 | -2.09 | -3.09 | -7.49 | -0.68 |
| CausalLM-CP 1.2B | 63.46 | 26.58 | 20.71 | 11.38 | 6.78 | 19.84 | 13.34 | 37.77 | 12.16 |
| | -1.30 | -9.79 | -7.15 | -5.28 | -2.89 | -8.20 | -5.00 | -10.02 | -1.04 |
| ObscuraCoder 2.8B | 65.42 | 45.29 | 35.97 | 20.11 | 14.77 | 48.78 | 23.55 | 61.93 | 14.15 |
| DOBF 2.8B | 65.37 | 37.92 | 29.10 | 16.69 | 13.18 | 43.65 | 20.32 | 51.27 | 13.39 |
| | -0.05 | -7.36 | -6.87 | -3.42 | -1.59 | -5.13 | -3.23 | -10.66 | -0.76 |
| CausalLM-CP 2.8B | 63.94 | 35.65 | 26.86 | 15.85 | 10.68 | 33.74 | 17.92 | 47.73 | 13.24 |
| | -1.48 | -9.64 | -9.11 | -4.26 | -4.09 | -15.04 | -5.63 | -14.20 | -0.91 |

Table 3: Results table for the comparative ablations of ObscuraCoder compared to a comparatively pre-trained DOBF model and a comparatively pre-trained CausalLM model, continually pre-trained on 8B tokens (CausalLM-CP) using obfuscation-grounded translation.

**Does Post-Hoc Obfuscation Training Suffice?** Finally, we test the effectiveness of our obfuscation translation training when it is post-hoc applied on an already pre-trained causal Code-LM. To this end, we subsample 8B tokens of translation pairs from ObscuraX and continue pre-training our causal LMs on obfuscation translation. The resulting models, dubbed CausalLM-CP, are compared against ObscuraCoder in Table 3. Unfortunately, we find that post-hoc obfuscation training does not confer performance gains to causally pre-trained LMs. This points to the importance of early obfuscation-grounding of code representations.

## 7 CONCLUSION

This work investigates the effects of obfuscation-grounding Code-LMs' representations across multiple programming languages. We create ObscuraX, an ≈119B-token source-code-to-obfuscated-code parallel dataset containing ≈55M training instances across seven languages. We then pre-train ObscuraCoder, a suite of models ranging from 255M to 2.8B parameters in size, on a ≈272B-token corpus that samples from ObscuraX and demonstrate that obfuscation grounding leads to substantial gains in syntactic and semantic understanding of code. We further uncover broader benefits of adding obfuscation translation to CodeLMs' pre-training mix: improved library-oriented code generation, multilingual code completion and summarization. Our results render obfuscation-based code translation a viable objective for bypassing the code data bottleneck. We hope our promising results catalyze the incorporation of obfuscated code into the pre-training recipes of frontier Code-LMs.

---

[11]While this leads to fewer gradient back-propagations for the DOBF model, we maintain that this is an inherent disadvantage that training with this objective in the current data-constrained regime incurs.

[12]The mainline GPTNeoX library does not support custom loss masking. For DOBF training, we employ a hitherto un-merged pull request containing the feature: ⤷ EleutherAI/gpt-neox/pull/1240

ACKNOWLEDGEMENTS

The work benefited from the Hessian Ministry of Higher Education and the Research and the Arts (HMWK) cluster project "The Third Wave of AI" as well as from the HMWK and BMBF joint support of the National Research Center for Applied Cybersecurity Center ATHENE. Additionally, the work acknowledges the support of Huawei Technologies (Ireland) Co., Ltd. Finally, the pre-training jobs were supported by a compute grant at the "FortyTwo" AI SuperPOD as part of the Hessian.AI Innovation Lab (funded by the Hessian Ministry for Digital Strategy and Innovation), grant no. S-DIW04/0013/003) and the hessian.AISC Service Center (funded by the Federal Ministry of Education and Research, BMBF, grant No 01IS22091).

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

## A  MODEL ARCHITECTURE & TRAINING DETAILS

| Attribute | ObscuraCoder 255M | ObscuraCoder 491M | ObscuraCoder 1.2B | ObscuraCoder 2.8B |
|---|---|---|---|---|
| **Training Attributes** | | | | |
| Scheduler Type | | | Cosine | |
| Scheduler Warmup Prop. | | | 0.05 | |
| Optimizer Type | | | AdamW | |
| Peak LR | | | 5e-4 | |
| Terminal LR | | | 2.5e-5 | |
| Beta | | | {0.9, 0.95} | |
| Epsilon | | | 1e-8 | |
| Gradient Clipping | | | 1.0 | |
| Weight Decay | | | 0.1 | |
| Deepspeed variant | | | Stage-2 | |
| Model Datatype | | | bfloat16 | |
| Softmax Datatype | | | float32 | |
| FP32 AllReduce | | | True | |
| Pipeline Parallel | | | False | |
| Activation Checkpointing | | | False | |
| Global Batch Size | | | 256 | |
| Training Steps | | | 520000 | |
| Contiguous Gradients | | | True | |
| Round Robin Gradients | | | True | |
| Scattered Reduce | | | True | |
| Partitioned Allgather | | | True | |
| **Architecture Attributes** | | | | |
| Norm Type | | | RMS | |
| Norm Epsilon | | | 1e-6 | |
| Pos. Embed. Type | | | Rotary | |
| Pos. Embed. Period | | | 1000000 | |
| Pos. Embed. Prop. | | | 1.0 | |
| Sequence Length | | | 2048 | |
| MLP Type | | | Llama | |
| Activation Function | | | SiLU | |
| Weight Tying | | | False | |
| Attention Variant | | | Flash Attention-2 | |
| Tokenizer Variant | | | Byte-Level BPE | |
| Tokenizer Vocab. Size | | | 49152 | |
| QK Norm | | | False | |
| Layer Count | 12 | 12 | 20 | 22 |
| QKV Heads Count | 8 | 12 | 16 | 24 |
| Hidden Size | 1024 | 1536 | 2048 | 3072 |
| Intermediate Size | 2816 | 4096 | 5632 | 8192 |
| Non Embed. Params. | 204M | 415M | 1128M | 2643M |
| Total Params. | 255M | 491M | 1229M | 2794M |

Table 4: Training and architectural attributes of the `ObscuraCoder` suite of models.

## B  DATASET DETAILS

### B.1  OBSCURAX COLLECTION

| Language | Sample Count | Token Count | | |
|---|---|---|---|---|
| | | Original | Obfuscated | Total |
| C | 5,986,186 | 11,619,163,328 | 10,962,297,717 | 22,581,461,045 |
| C++ | 5,374,108 | 9,470,872,459 | 8,961,976,794 | 18,432,849,253 |
| Go | 5,188,074 | 5,063,743,140 | 4,903,380,220 | 9,967,123,360 |
| Java | 18,523,273 | 18,064,491,641 | 17,411,757,778 | 35,476,249,419 |
| Python | 13,600,833 | 11,183,172,044 | 10,718,436,155 | 21,901,608,199 |
| Rust | 1,263,892 | 1,533,690,227 | 1,485,340,352 | 3,019,030,579 |
| Typescript | 5,403,241 | 3,855,020,996 | 3,726,795,270 | 7,581,816,266 |
| Total: | 55,339,607 | 60,790,153,835 | 58,169,984,286 | 118,960,138,121 |

Table 5: Language-wise sample and token count breakdown of the `ObscuraX` dataset.

#### B.1.1  DATASET LIMITATIONS & FAILED COLLECTION ATTEMPTS

**Limitations.** `ObscuraX` is collected with the stated objective of allowing Code-LMs to better disentangle code syntax and semantics using obfuscation as a semantic-preserving augmentation. However, there exist some corner cases beyond the import obfuscation alluded to in Section 3, where the correctness of the original code is not preserved.

One such scenario is wildcard or global module-level imports, where our obfuscator treats the imported members no differently from user-defined ones. This can affect correctness in cases such as

\* imports in Python or when header-only libraries are used in languages like C++. In practice, we find the occurrence of these to be limited and hence do not tailor our obfuscator for them. Another scenario is the potential obfuscation of standard macros in languages like Rust and C. These are pre-defined by the language standard and encapsulate platform-level functionality. However, their use is exceedingly rare, and we avoid any special handling of these structures.

**Failed collection attempts.** Before building a custom obfuscator, we attempted to re-purpose existing open-source semantic highlighting toolchains for obfuscation. We detail two failed efforts below:

- **git** `github/semantic`: The semantic highlighting package made public by GitHub was a natural first attempt. However, the framework seeks to map elements in all programming languages to a closed set of Haskell constructs, which can be limiting. Furthermore, it does not expose a Tree-sitter-like query layer for targeted extraction of information.
- **git** `microsoft/language-server-protocol`: We attempted to instrument the semantic highlighting mechanism of popular IDEs for obfuscation, by directly extracting all spans relating to specific identifiers. However, in practice, the language servers of many languages are immature and slow. We were further hamstrung by the fact that most language servers work reliably only at the repository level, which was at odds with the file-level sourcing of `ObscuraX`.

## B.2    PRE-TRAINING CORPUS COLLECTION

**`filtered-code-text` collection.** The `filtered-code-text` collection consists of a text-only split consisting of roughly 105B tokens and a code-adjacent split comprised of a further 15B tokens. The text-only split's sourcing is biased towards technical, academic and educational content. We list the constituent HuggingFace sources of the text-only split along with the at-source filtering procedures (if any) below:

- 🤗 `allenai/c4`: We subset for the `realnewslike` split.
- 🤗 `Skylion007/openwebtext`
- 🤗 `allenai/peS2o`: We use regex-based removal of citation text and bibliography information.
- 🤗 `datajuicer/redpajama-book-refined-by-data-juicer`
- 🤗 `datajuicer/redpajama-pile-stackexchange-refined-by-data-juicer`
- 🤗 `datajuicer/the-pile-uspto-refined-by-data-juicer`
- 🤗 `datajuicer/redpajama-wiki-refined-by-data-juicer`
- 🤗 `datajuicer/the-pile-nih-refined-by-data-juicer`
- 🤗 `SciPhi/textbooks-are-all-you-need-lite`
- 🤗 `wentingzhao/math-textbooks`
- 🤗 `airtrain-ai/fineweb-edu-fortified`: We further bias for educational content by selecting samples with an educational `score` greater than 3.5 and a repeat `count` greater than 2.

The code-adjacent split focuses on organic and synthetic data sources where text and code appear interspersed, with the text fulfilling a descriptive or pedagogical role. The following lists the constituent HuggingFace sources of this split along with the at-source filtering procedures (if any):

- 🤗 `vikp/textbook_quality_programming`
- 🤗 `bigcode/stack-exchange-preferences-20230914-clean`: We select QA pairs where the answer has been accepted by the community and the answerer has a reputation of $\geq 3$. We also use QA pairs from only the following sub-domains: `stackoverflow`, `serverfault`, `askubuntu`, `superuser`, and `stackexchange`.
- 🤗 `vikp/code_with_explanations`
- 🤗 `nampdn-ai/devdocs.io`
- 🤗 `open-phi/programming _books_llama`
- 🤗 `SivilTaram/starcoder2-documentation`
- 🤗 `bigcode/coding_tutorials`

- 🤗 `bigcode/commitpackft`
- 🤗 `bigcode/the-stack-github-issues`

The two splits are merged, shuffled and subsequently subjected to the following steps of filtering:

- We follow recent work (Marion et al., 2023) and prune our corpus using a perplexity filter to improve pre-training efficiency. We leverage a KenLM (Heafield, 2011) model trained on the EN OSCAR (Abadji et al., 2022) corpus and discard documents with a length-normalized perplexity of more than 325 and less than 7.

- We follow established practice (Penedo et al., 2023) and discard samples with any <=4-gram whose repeats constitute greater than 15% of the respective n-gram count.

- We remove samples whose English language score is lower than 99%. We employ the Lingua [13] language detector to make this determination.

- Finally, we perform an aggressive MinHash (Broder, 1997) de-duplication using a shingle size of 8 and a similarity threshold of 0.5. This allows us to efficiently approximate frontier model performance with constrained compute (Lee et al., 2022).

**`filtered-source-code` collection.** The `filtered-source-code` collection is the result of pruning an already filtered version [14] of the Stack (Kocetkov et al., 2023). We subset for the seven language splits, which we evaluate on — C, C++, Go, Java, Python, Rust, and TypeScript — and supplement it with the Markdown split. On the resultant data, we further apply the following filters:

- For files forked more than 25 times, we retain them if the average line length is less than 140, the maximum line length is less than 500, and the alphanumeric fraction is more than 25%.

- For files forked between 10 and 25 times, we retain them if the average line length is less than 120, the maximum line length is less than 200, and the alphanumeric fraction is more than 35%.

- For files forked less than 10 times, we retain them if the average line length is less than 100, the maximum line length is less than 200, and the alphanumeric fraction is more than 40%.

- We only retain samples from conventionally used file extensions and drop samples from valid but uncommon extensions.

- Seeking to avoid the deleterious effects of near-duplicate data on Code-LMs (Allamanis, 2019), we subject the resultant data to an aggressive MinHash (Broder, 1997) de-duplication, using a shingle size of 20 and a similarity threshold of 0.75.

| Language | Chosen Extensions | Sample Count | Token Count |
|---|---|---|---|
| C | {.c, .h} | 8,149,577 | 14,381,796,257 |
| C++ | {.cpp, .c++, .cc, .cxx, .h++, .hh, .hpp, .hxx} | 5,923,165 | 10,912,556,820 |
| Go | {.go} | 4,507,348 | 9,967,123,360 |
| Java | {.java} | 19,324,891 | 16,249,736,121 |
| Markdown | {.md, .markdown} | 12,623,416 | 5,614,520,807 |
| Python | {.py} | 12,304,123 | 12,868,526,213 |
| Rust | {.rs} | 1,314,569 | 1,932,035,717 |
| Typescript | {.ts, .tsx} | 8,207,133 | 4,437,582,117 |
| Total: | | 72,354,222 | 76,363,877,412 |

Table 6: Language-wise chosen extensions along with sample and token count breakdown of the `filtered-source-code` collection used to construct `ObscuraCoder`, DOBF, and Causal LM pre-training corpora.

### B.3 OBFUSCATED CODE EXAMPLES

For the reader's reference, we provide complete `ObscuraX` samples in all supported languages along with the accompanying metadata.

---

[13] git `pemistahl/lingua-py`
[14] 🤗 `bigcode/the-stack-dedup`

**Original Python Code**

```python
from __future__ import print_function, division, absolute_import
import sys, os.path

_importlist = "VisitorBase parse check".split()

_cd = os.path.join(os.path.dirname(__file__))
version_specific_path = None
common_path = os.path.join(_cd, 'common')

def _get_asdl_depending_on_version():

    global version_specific_path
    major, minor = sys.version_info[0], sys.version_info[1]
    prefix = __name__.rsplit('.', 1)
    base = (prefix[0] + '.') if len(prefix) > 1 else ''
    dname = 'py%d_%d' % (major, minor)
    version_specific_path = os.path.join(_cd, dname)

    use_abs_import = 0

    mname = base + dname + '.asdl'
    try:
        mod = __import__(mname, fromlist=_importlist, level=use_abs_import)
    except ImportError:
        dname = 'common'
        mname = base + dname + '.asdl'
        mod = __import__(mname, fromlist=_importlist)
    for i in _importlist:
        globals()[i] = getattr(mod, i)
```

**Obfuscated Python Code**

```python
from __future__ import IMPORT_0, IMPORT_1, IMPORT_2
import sys, os.IMPORT_3

VAR_0 = "VisitorBase parse check".FUNC_0()

VAR_1 = os.IMPORT_3.FUNC_1(os.IMPORT_3.FUNC_2(VAR_2))
VAR_3 = None
common_path = os.IMPORT_3.FUNC_1(VAR_1, 'common')

def _get_asdl_depending_on_version():

    global VAR_3
    VAR_4, VAR_5 = sys.VAR_6[0], sys.VAR_6[1]
    VAR_7 = VAR_8.FUNC_3('.', 1)
    base = (VAR_7[0] + '.') if FUNC_4(VAR_7) > 1 else ''
    VAR_9 = 'py%d_%d' % (VAR_4, VAR_5)
    VAR_3 = os.IMPORT_3.FUNC_1(VAR_1, VAR_9)

    VAR_10 = 0

    VAR_11 = base + VAR_9 + '.asdl'
    try:
        VAR_12 = FUNC_5(VAR_11, VAR_13=VAR_0, VAR_14=VAR_10)
    except VAR_15:
        VAR_9 = 'common'
        VAR_11 = base + VAR_9 + '.asdl'
        VAR_12 = FUNC_5(VAR_11, VAR_13=VAR_0)
    for VAR_16 in VAR_0:
        FUNC_6()[VAR_16] = FUNC_7(VAR_12, VAR_16)
```

**Metadata**

```
Obfuscation proportion: 85.6%
```
- - - - - - - - - - - - - - - - - - - - - - - - - - - - - - - - - - - - - - - - - - - - - -
```
'IMPORT_0': 'print_function', 'IMPORT_1': 'division', 'IMPORT_2': 'absolute_import', 'IMPORT_3': 'path', 'VAR_0': '_importlist', 'FUNC_0': 'split', 'VAR_1': '_cd', 'FUNC_1': 'join', 'FUNC_2': 'dirname', 'VAR_2': '__file__', 'VAR_3': 'version_specific_path', 'VAR_4': 'major', 'VAR_5': 'minor', 'VAR_6': 'version_info', 'VAR_7': 'prefix', 'VAR_8': '__name__', 'FUNC_3': 'rsplit', 'FUNC_4': 'len', 'VAR_9': 'dname', 'VAR_10': 'use_abs_import', 'VAR_11': 'mname', 'VAR_12': 'mod', 'FUNC_5': '__import__', 'VAR_13': 'fromlist', 'VAR_14': 'level', 'VAR_15': 'ImportError', 'VAR_16': 'i', 'FUNC_6': 'globals', 'FUNC_7': 'getattr', 'FUNC_8': 'load', 'VAR_17': 'filename', 'VAR_18': 'srcfile', 'FUNC_9': 'exists', 'VAR_19': 'asdl', 'FUNC_10': 'parse', 'FUNC_11': 'check'.
```

Figure 3: An example of original and obfuscated Python code in `ObscuraX` along with accompanying metadata stating the obfuscation proportion and the identifier map.

**Original Java Code**

```java
package com.dji.GSDemo.GoogleMap;

import android.os.Bundle;
import android.support.v7.app.AppCompatActivity;
import android.content.Intent;
import android.widget.TextView;

public class showDetail extends AppCompatActivity {
    @Override
    protected void onCreate(Bundle savedInstanceState) {
        super.onCreate(savedInstanceState);
        setContentView(R.layout.show_details);

        TextView result = (TextView) findViewById(R.id.textView2);
        TextView resultData = (TextView) findViewById(R.id.textView3);
        Intent in = getIntent();
        result.setText(in.getStringExtra("name"));
        resultData.setText(in.getStringExtra("data"));
    }
}
```

**Obfuscated Java Code**

```java
package IMPORT_0.dji.GSDemo.GoogleMap;

import android.os.Bundle;
import android.IMPORT_1.v7.app.AppCompatActivity;
import android.content.IMPORT_2;
import android.widget.TextView;

public class showDetail extends AppCompatActivity {
    @Override
    protected void onCreate(Bundle savedInstanceState) {
        super.onCreate(savedInstanceState);
        setContentView(R.layout.show_details);

        TextView result = (TextView) findViewById(R.id.textView2);
        TextView resultData = (TextView) findViewById(R.id.textView3);
        IMPORT_2 VAR_0 = getIntent();
        result.FUNC_0(VAR_0.getStringExtra("name"));
        resultData.FUNC_0(VAR_0.getStringExtra("data"));
    }
}
```

**Metadata**

```
Obfuscation proportion: 12.0%
```
- - - - - - - - - - - - - - - - - - - - - - - - - - - - - - - - - - - - - - - - - - - - - -
```
'IMPORT_0': 'com', 'IMPORT_1': 'support', 'IMPORT_2': 'Intent', 'VAR_0': 'in', 'FUNC_0': 'setText'.
```

Figure 4: An example of original and obfuscated Java code in `ObscuraX` along with accompanying metadata stating the obfuscation proportion and the identifier map.

**Original C++ Code**

```cpp
#include <iostream>
#include <stdio.h>
#include <string.h>
using namespace std;
int n,m;
int data;
int tree[1005];
char str[10][10];
int lowbit(int i)
{
    return i&(-i);
}
void add(int i,int x)
{
    while(i<=n)
    {
        tree[i]+=x;
        i+=lowbit(i);
    }
}
int sum(int i)
{
    int ans=0;
    while(i>0)
    {
        ans+=tree[i];
        i-=lowbit(i);
    }
    return ans;
}
```

**Obfuscated C++ Code**

```cpp
#include <iostream>
#include <IMPORT_0>
#include <string.h>
using namespace std;
int VAR_0,VAR_1;
int data;
int tree[1005];
char str[10][10];
int lowbit(int VAR_2)
{
    return VAR_2&(-VAR_2);
}
void add(int VAR_2,int VAR_3)
{
    while(VAR_2<=VAR_0)
    {
        tree[VAR_2]+=VAR_3;
        VAR_2+=lowbit(VAR_2);
    }
}
int sum(int VAR_2)
{
    int ans=0;
    while(VAR_2>0)
    {
        ans+=tree[VAR_2];
        VAR_2-=lowbit(VAR_2);
    }
    return ans;
}
```

**Metadata**

```
Obfuscation proportion: 29.9%
```

```
'IMPORT_0': 'stdio.h', 'VAR_0': 'n', 'VAR_1': 'm', 'VAR_2': 'i', 'VAR_3': 'x'.
```

Figure 5: An example of original and obfuscated C++ code in `ObscuraX` along with accompanying metadata stating the obfuscation proportion and the identifier map.

**Original C Code**

```c
void dshot_dma_start() {
  uint32_t time = time_micros();
  while ((dshot_dma_phase != 0 || spi_dma_is_ready(SPI_PORT1) == 0) && (
  time_micros() - time) < state.looptime * 1e6f)
    ;
  if (dshot_dma_phase != 0 || spi_dma_is_ready(SPI_PORT1) == 0)
    return;
  for (uint8_t i = 0; i < 16; i++) {
    motor_data_portA[i] = 0;
    motor_data_portB[i] = 0;
    motor_data_portC[i] = 0;
#define MOTOR_PIN(port, pin, pin_af, timer, timer_channel)     \
  if (!(dshot_packet[MOTOR_PIN_IDENT(port, pin)] & 0x8000)) { \
    if (GPIO##port == GPIOA)                                   \
      motor_data_portA[i] |= (LL_GPIO_PIN_##pin << 16);        \
    else if (GPIO##port == GPIOB)                             \
      motor_data_portB[i] |= (LL_GPIO_PIN_##pin << 16);        \
    else if (GPIO##port == GPIOC)                             \
      motor_data_portC[i] |= (LL_GPIO_PIN_##pin << 16);        \
  }
    MOTOR_PINS
#undef MOTOR_PIN
    dshot_packet[0] <<= 1;
    dshot_packet[1] <<= 1;
    dshot_packet[2] <<= 1;
    dshot_packet[3] <<= 1;
  }
  for (int i = 1, j = 0; i < 48 && j < 16; i += 3, j++) {
    portA_buffer[i] = motor_data_portA[j];
    portB_buffer[i] = motor_data_portB[j];
    portC_buffer[i] = motor_data_portC[j];
  }
  dshot_dma_phase = DSHOT_PORT_COUNT;
  TIM1->ARR = DSHOT_BIT_TIME;
  TIM1->CCR1 = 0;
  TIM1->CCR2 = DSHOT_T0H_TIME;
  TIM1->CCR3 = DSHOT_T1H_TIME;
  if (DSHOT_GPIO_A == 1)
    dshot_dma_portA();
  else if (DSHOT_GPIO_B == 1)
    dshot_dma_portB();
  else if (DSHOT_GPIO_C == 1)
    dshot_dma_portC();
}
```

**Obfuscated C Code**

```c
void FUNC_0() {
  uint32_t VAR_0 = FUNC_1();
  while ((VAR_1 != 0 || FUNC_2(VAR_2) == 0) && (FUNC_1() - VAR_0) < VAR_3.
  VAR_4 * 1e6f)
    ;
  if (VAR_1 != 0 || FUNC_2(VAR_2) == 0)
    return;
  for (uint8_t VAR_5 = 0; VAR_5 < 16; VAR_5++) {
    VAR_6[VAR_5] = 0;
    VAR_7[VAR_5] = 0;
    VAR_8[VAR_5] = 0;
#define FUNC_3(VAR_9, VAR_10, VAR_11, VAR_12, VAR_13)     \
  if (!(dshot_packet[MOTOR_PIN_IDENT(port, pin)] & 0x8000)) { \
    if (GPIO##port == GPIOA)                                   \
      motor_data_portA[i] |= (LL_GPIO_PIN_##pin << 16);        \
    else if (GPIO##port == GPIOB)                             \
      motor_data_portB[i] |= (LL_GPIO_PIN_##pin << 16);        \
    else if (GPIO##port == GPIOC)                             \
      motor_data_portC[i] |= (LL_GPIO_PIN_##pin << 16);        \
  }
    CLASS_0
#undef MOTOR_PIN
    VAR_14[0] <<= 1;
    VAR_14[1] <<= 1;
    VAR_14[2] <<= 1;
    VAR_14[3] <<= 1;
  }
  for (int VAR_5 = 1, VAR_15 = 0; VAR_5 < 48 && VAR_15 < 16; VAR_5 += 3,
  VAR_15++) {
    VAR_16[VAR_5] = VAR_6[VAR_15];
    VAR_17[VAR_5] = VAR_7[VAR_15];
    VAR_18[VAR_5] = VAR_8[VAR_15];
  }
  VAR_1 = VAR_19;
  VAR_20->VAR_21 = VAR_22;
  VAR_20->VAR_23 = 0;
  VAR_20->VAR_24 = DSHOT_T0H_TIME;
  VAR_20->VAR_25 = DSHOT_T1H_TIME;
  if (VAR_26 == 1)
    FUNC_4();
  else if (VAR_27 == 1)
    FUNC_5();
  else if (VAR_28 == 1)
    FUNC_6();
}
```

**Metadata**

```
Obfuscation proportion: 82.1%
```

- - - - - - - - - - - - - - - - - - - - - - - - - - - - - - - - - - - - - - - - - - - - - - - - - - - -

```
'FUNC_0': 'dshot_dma_start', 'VAR_0': 'time', 'FUNC_1': 'time_micros', 'VAR_1': 'dshot_dma_phase', 'FUNC_2': 'spi_dma_is_ready', 'VAR_2': 'SPI_PORT1', 'VAR_3'
: 'state', 'VAR_4': 'looptime', 'VAR_5': 'i', 'VAR_6': 'motor_data_portA', 'VAR_7': 'motor_data_portB', 'VAR_8': 'motor_data_portC', 'FUNC_3': 'MOTOR_PIN', '
VAR_9': 'port', 'VAR_10': 'pin', 'VAR_11': 'pin_af', 'VAR_12': 'timer', 'VAR_13': 'timer_channel', 'CLASS_0': 'MOTOR_PINS', 'VAR_14': 'dshot_packet', 'VAR_15'
: 'j', 'VAR_16': 'portA_buffer', 'VAR_17': 'portB_buffer', 'VAR_18': 'portC_buffer', 'VAR_19': 'DSHOT_PORT_COUNT', 'VAR_20': 'TIM1', 'VAR_21': 'ARR', 'VAR_22'
: 'DSHOT_BIT_TIME', 'VAR_23': 'CCR1', 'VAR_24': 'CCR2', 'VAR_25': 'CCR3', 'VAR_26': 'DSHOT_GPIO_A', 'FUNC_4': 'dshot_dma_portA', 'VAR_27': 'DSHOT_GPIO_B', '
FUNC_5': 'dshot_dma_portB', 'VAR_28': 'DSHOT_GPIO_C', 'FUNC_6': 'dshot_dma_portC'.
```

Figure 6: An example of original and obfuscated C code in `ObscuraX` along with accompanying metadata stating the obfuscation proportion and the identifier map.

<div>

**Original Rust Code**

```rust
/// Generate an encrypted header
/// for a resource encrypted using an hybrid crypto scheme.
///
/// A random symmetric key is generated for the specified symmetric scheme,
/// encrypted using the public key of the ABE scheme and policy attributes
/// then pre-pended to the symmetrically encrypted metadata
pub fn encrypt_hybrid_header<A, S>(
    policy: &Policy,
    public_key: &A::MasterPublicKey,
    attributes: &[Attribute],
    meta_data: Option<Metadata>,
) -> Result<EncryptedHeader, FormatErr>
where
    A: AbeScheme + std::marker::Sync + std::marker::Send,
    S: SymmetricCrypto,
{
    let engine = Engine::<A>::new();
    let (sk_bytes, encrypted_sk) =
        engine.generate_symmetric_key(policy, public_key, attributes, S::Key
        ::LENGTH)?;
    let symmetric_key = S::Key::try_from_bytes(sk_bytes)?;
    // convert to bytes
    // ..size
    let mut header_bytes = u32_len(&encrypted_sk)?.to_vec();
    // ...bytes
    header_bytes.extend(&encrypted_sk);
    if let Some(meta) = meta_data {
        // Nonce
        let nonce = S::Nonce::new(&mut CsRng::new());
        header_bytes.extend(nonce.to_bytes());
        // Encrypted metadata
        let encrypted_metadata = S::encrypt(&symmetric_key, &meta.to_bytes()
        ?, &nonce, None)?;
        // ... size
        header_bytes.extend(u32_len(&encrypted_metadata)?);
        // ... bytes
        header_bytes.extend(encrypted_metadata);
    }
    Ok(EncryptedHeader {
        symmetric_key,
        encrypted_header_bytes: header_bytes,
    })
}
```

</div>

<div>

**Obfuscated Rust Code**

```rust
/// Generate an encrypted header
/// for a resource encrypted using an hybrid crypto scheme.
///
/// A random symmetric key is generated for the specified symmetric scheme,
/// encrypted using the public key of the ABE scheme and policy attributes
/// then pre-pended to the symmetrically encrypted metadata
pub fn encrypt_hybrid_header<A, S>(
    policy: &Policy,
    public_key: &A::MasterPublicKey,
    attributes: &[Attribute],
    meta_data: Option<Metadata>,
) -> Result<EncryptedHeader, FormatErr>
where
    A: AbeScheme + std::marker::Sync + std::marker::Send,
    S: SymmetricCrypto,
{
    let engine = Engine::<A>::new();
    let (sk_bytes, encrypted_sk) =
        engine.generate_symmetric_key(policy, public_key, attributes, S::Key
        ::LENGTH)?;
    let symmetric_key = S::Key::try_from_bytes(sk_bytes)?;
    // convert to bytes
    // ..size
    let mut header_bytes = u32_len(&encrypted_sk)?.to_vec();
    // ...bytes
    header_bytes.extend(&encrypted_sk);
    if let Some(meta) = meta_data {
        // Nonce
        let nonce = S::Nonce::new(&mut CsRng::new());
        header_bytes.extend(nonce.FUNC_0());
        // Encrypted metadata
        let encrypted_metadata = S::encrypt(&symmetric_key, &meta.FUNC_0()?,
         &nonce, None)?;
        // ... size
        header_bytes.extend(u32_len(&encrypted_metadata)?);
        // ... bytes
        header_bytes.extend(encrypted_metadata);
    }
    Ok(EncryptedHeader {
        symmetric_key,
        VAR_0: header_bytes,
    })
}
```

</div>

<div>

**Metadata**

```
Obfuscation proportion: 2.6%
```

- - - - - - - - - - - - - - - - - - - - - - - - - - - - - - - - - - - - - -

```
'FUNC_0': 'to_bytes', 'VAR_0': 'encrypted_header_bytes'
```

</div>

Figure 7: An example of original and obfuscated Rust code in `ObscuraX` along with accompanying metadata stating the obfuscation proportion and the identifier map.



**Original Typescript Code**

```typescript
export default (data: any[], period: number = 21) => {
  let position = "BULL"

  // console.log(data)
  // get data from the end of the dataset backward (positive to negative
  number)
  const dataSet = data.slice(-Math.abs(period))

  // calculate simple moving average for the period
  const sma = dataSet.map((item: any) => item.close).reduce((prev: number,
  curr: number) => prev + curr, 0) / period

  // Bull or Bear?
  // if the last price is above the SMA = Bull
  // if the last price is below the SMA = Bear
  const lastDataPoint = data.slice(-1)[0]
  // console.log(101, lastDataPoint)

  const { startTime, close } = lastDataPoint

  if (close < sma) {
    position = "BEAR"
  }

  return { startTime, sma, position }
}
```

</div>



**Obfuscated Typescript Code**

```typescript
export default (data: any[], period: number = 21) => {
  let position = "BULL"

  // console.log(data)
  // get data from the end of the dataset backward (positive to negative
  number)
  const VAR_0 = data.slice(-Math.abs(period))

  // calculate simple moving average for the period
  const VAR_1 = VAR_0.FUNC_0((item: any) => item.VAR_2).reduce((prev: number
  , VAR_3: number) => prev + VAR_3, 0) / period

  // Bull or Bear?
  // if the last price is above the SMA = Bull
  // if the last price is below the SMA = Bear
  const lastDataPoint = data.slice(-1)[0]
  // console.log(101, lastDataPoint)

  const { startTime, close } = lastDataPoint

  if (VAR_2 < VAR_1) {
    position = "BEAR"
  }

  return { startTime, VAR_1, position }
}
```

</div>

**Metadata**

```
Obfuscation proportion: 25.1%
- - - - - - - - - - - - - - - - - - - - - - - - - - - - - - - - - - - - - - - - - - - - - - - - - - - - - -
'VAR_0': 'dataSet', 'VAR_1': 'sma', 'FUNC_0': 'map', 'VAR_2': 'close', 'VAR_3': 'curr'.
```

Figure 8: An example of original and obfuscated TypeScript code in ObscuraX along with accompanying metadata stating the obfuscation proportion and the identifier map.

Figure 9: An example of original and obfuscated Go code in `ObscuraX` along with accompanying metadata stating the obfuscation proportion and the identifier map.

# C    DETAILED RESULTS

For completeness, we detail the split and language-wise performance of the models on all tasks (where applicable) discussed in Section 5.

| ⌨ Model | 🗨 Doc to Comments | ↵ Newline After Code | ▣ Newline After Doc | ⊡ Newline Random | ↕ Line Split | ⇥ Tab Indent |
|---|---|---|---|---|---|---|
| CausalLM 255M | 9.75 | 12.19 | 10.99 | 10.99 | 11.51 | 12.77 |
| ObscuraCoder 255M | 11.58 **+1.83** | 15.91 **+3.72** | 14.98 **+3.99** | 12.91 **+1.92** | 14.97 **+3.46** | 14.67 **+1.90** |
| CausalLM 491M | 12.19 | 15.88 | 16.46 | 13.41 | 14.02 | 18.29 |
| ObscuraCoder 491M | 21.34 **+9.15** | 26.21 **+10.33** | 26.82 **+10.36** | 21.95 **+8.54** | 22.56 **+8.54** | 24.39 **+6.10** |
| CausalLM 1.2B | 22.89 | 29.55 | 29.55 | 26.87 | 27.43 | 29.59 |
| ObscuraCoder 1.2B | 33.56 **+10.67** | 36.78 **+7.23** | 35.36 **+5.81** | 36.97 **+10.10** | 36.46 **+9.03** | 39.11 **+9.52** |
| CausalLM 2.8B | 30.78 | 35.11 | 36.93 | 34.01 | 35.67 | 35.67 |
| ObscuraCoder 2.8B | 38.93 **+8.15** | 46.71 **+11.60** | 48.36 **+11.43** | 44.74 **+10.73** | 48.06 **+12.39** | 44.91 **+9.24** |
| StarCoderBase 1B | 25.77 | 31.67 | 31.67 | 27.43 | 28.65 | 28.65 |
| DeepSeekCoder 1.3B | 41.36 | 52.94 | 46.88 | 50.68 | 44.91 | 51.47 |
| StarCoderBase 3B | 35.36 | 37.35 | 39.63 | 35.67 | 37.07 | 38.17 |
| StarCoder2 3B | 49.12 | 60.31 | 49.73 | 51.94 | 54.58 | 57.21 |
| StableCode 3B | 44.89 | 50.79 | 55.48 | 48.79 | 50.28 | 50.28 |
| CodeGemma 2B | 9.56 | 11.71 | 12.97 | 9.14 | 10.53 | 7.92 |
| Phi-2 | 58.25 | 59.34 | 68.12 | 61.46 | 49.12 | 63.63 |

Table 7: ReCode Format `pass@1` comparison between `ObscuraCoder` and comparable Causal LM models, along with frontier models for context.

| ⌨ Model | ♔ Dead Code Insert | ♺ For While Transform | ⇄ Operand Swap | 🖩 Var Renaming CB | 🖩 Var Renaming Naive | 🖩 Var Renaming RN |
|---|---|---|---|---|---|---|
| CausalLM 255M | 2.63 | 10.43 | 12.84 | 12.43 | 6.11 | 9.45 |
| ObscuraCoder 255M | 3.66 **+1.03** | 15.48 **+5.05** | 15.48 **+2.64** | 15.87 **+3.44** | 8.37 **+2.26** | 11.96 **+2.51** |
| CausalLM 491M | 4.87 | 13.74 | 16.46 | 13.74 | 7.92 | 11.06 |
| ObscuraCoder 491M | 8.53 **+3.66** | 26.37 **+12.63** | 25.14 **+8.68** | 26.99 **+13.25** | 14.21 **+6.29** | 18.97 **+7.91** |
| CausalLM 1.2B | 8.12 | 25.87 | 27.11 | 26.96 | 14.77 | 23.21 |
| ObscuraCoder 1.2B | 10.82 **+2.70** | 34.77 **+8.90** | 37.91 **+10.80** | 34.56 **+7.60** | 19.96 **+5.19** | 29.13 **+5.92** |
| CausalLM 2.8B | 10.22 | 34.13 | 36.15 | 35.97 | 14.98 | 25.76 |
| ObscuraCoder 2.8B | 17.03 **+6.81** | 44.88 **+10.75** | 47.56 **+11.41** | 44.98 **+9.01** | 23.25 **+8.27** | 38.13 **+12.37** |
| StarCoderBase 1B | 8.53 | 32.19 | 29.12 | 32.19 | 28.77 | 24.70 |
| DeepSeekCoder 1.3B | 17.19 | 51.78 | 50.24 | 55.06 | 49.36 | 42.87 |
| StarCoderBase 3B | 12.87 | 25.59 | 37.97 | 38.67 | 35.79 | 30.61 |
| StarCoder2 3B | 16.14 | 53.38 | 57.44 | 59.14 | 50.85 | 48.78 |
| StableCode 3B | 14.13 | 52.77 | 56.01 | 49.64 | 49.06 | 39.97 |
| CodeGemma 2B | 6.79 | 12.12 | 11.67 | 11.49 | 9.54 | 6.11 |
| Phi-2 | 18.15 | 64.19 | 63.56 | 62.26 | 59.77 | 54.35 |

Table 8: ReCode Syntax `pass@1` comparison between `ObscuraCoder` and comparable Causal LM models, along with frontier models for context.

| ⌨ Model | ∿ Camel Case | ⇢ Butter Fingers | ⇅ Swap Characters | ✂ Change Character Case | ⬚ Inflectional Variation | ⬓ Synonym Substitution |
|---|---|---|---|---|---|---|
| CausalLM 255M | 4.26 | 1.97 | 2.03 | 2.78 | 2.89 | 4.82 |
| ObscuraCoder 255M | 5.74 **+1.48** | 4.26 **+2.29** | 4.78 **+2.75** | 3.14 **+0.36** | 5.63 **+2.74** | 6.13 **+1.31** |
| CausalLM 491M | 6.71 | 4.87 | 5.49 | 2.43 | 6.97 | 4.87 |
| ObscuraCoder 491M | 11.59 **+4.88** | 6.71 **+1.84** | 8.56 **+3.07** | 6.11 **+3.68** | 8.95 **+1.98** | 6.11 **+1.24** |
| CausalLM 1.2B | 15.49 | 11.55 | 10.16 | 8.74 | 11.69 | 10.16 |
| ObscuraCoder 1.2B | 20.08 **+4.59** | 16.21 **+4.66** | 17.17 **+7.01** | 13.44 **+4.70** | 17.17 **+5.48** | 15.89 **+5.73** |
| CausalLM 2.8B | 18.12 | 14.29 | 15.98 | 8.53 | 15.98 | 15.44 |
| ObscuraCoder 2.8B | 25.32 **+7.20** | 19.26 **+4.97** | 17.12 **+1.14** | 16.68 **+8.15** | 21.40 **+5.42** | 20.86 **+5.42** |
| StarCoderBase 1B | 15.07 | 13.85 | 13.09 | 13.41 | 14.19 | 13.41 |
| DeepSeekCoder 1.3B | 27.43 | 24.82 | 26.56 | 19.45 | 28.28 | 25.10 |
| StarCoderBase 3B | 20.34 | 17.24 | 16.89 | 15.01 | 16.56 | 18.91 |
| StarCoder2 3B | 30.65 | 26.57 | 23.61 | 18.11 | 25.39 | 23.17 |
| StableCode 3B | 28.97 | 23.48 | 23.11 | 19.35 | 26.88 | 23.11 |
| CodeGemma 2B | 8.53 | 5.67 | 7.93 | 4.87 | 8.48 | 6.78 |
| Phi-2 | 41.36 | 29.14 | 30.22 | 26.41 | 37.44 | 31.67 |

Table 9: ReCode Function `pass@1` comparison between `ObscuraCoder` and comparable Causal LM models, along with frontier models for context.

| Model | C++ | Java | Python | Rust | TypeScript |
|---|---|---|---|---|---|
| CausalLM 255M | 3.51 | 4.29 | 4.32 | 2.78 | 6.53 |
| ObscuraCoder 255M | 6.94 | 6.27 | 6.47 | 3.46 | 6.53 |
| | +3.43 | +1.98 | +2.15 | +0.68 | 0.00 |
| CausalLM 491M | 5.74 | 5.74 | 7.20 | 2.74 | 7.79 |
| ObscuraCoder 491M | 8.56 | 8.60 | 10.91 | 6.61 | 9.13 |
| | +2.82 | +2.86 | +3.71 | +3.87 | +1.34 |
| CausalLM 1.2M | 12.87 | 13.04 | 13.89 | 9.71 | 10.82 |
| ObscuraCoder 1.2B | 19.03 | 18.26 | 19.79 | 15.69 | 18.92 |
| | +6.16 | +5.22 | +5.90 | +5.98 | +8.10 |
| CausalLM 1.2M | 17.79 | 16.48 | 19.16 | 14.28 | 15.79 |
| ObscuraCoder 2.8B | 25.08 | 23.56 | 26.34 | 16.93 | 25.86 |
| | +7.29 | +7.08 | +7.18 | +2.65 | +10.07 |
| StarCoderBase 1B | 12.05 | 14.11 | 15.06 | 10.22 | 14.18 |
| DeepSeekCoder 1.3B | 29.88 | 29.11 | 28.87 | 18.69 | 27.52 |
| StarCoderBase 3B | 20.46 | 18.64 | 20.19 | 16.83 | 19.92 |
| StarCoder2 3B | 26.78 | 27.89 | 26.49 | 25.56 | 30.16 |
| StableCode 3B | 28.04 | 25.31 | 29.84 | 23.03 | 27.44 |
| CodeGemma 2B | 27.26 | 22.42 | 20.78 | 28.42 | 14.65 |
| Phi-2 | 21.75 | 20.88 | 49.92 | 6.87 | 11.94 |

Table 10: Multipl-E `pass@1` comparison between `ObscuraCoder` and comparable Causal LM models, along with frontier models for context.

| Model | C++ | Java | Python | Rust | TypeScript |
|---|---|---|---|---|---|
| CausalLM 255M | 7.34 | 7.62 | 9.45 | 4.08 | 7.15 |
| ObscuraCoder 255M | 9.75 | 10.06 | 13.95 | 5.59 | 8.95 |
| | +2.41 | +2.44 | +4.50 | +1.51 | +1.80 |
| CausalLM 491M | 10.41 | 10.38 | 13.81 | 7.59 | 11.02 |
| ObscuraCoder 491M | 14.29 | 13.24 | 17.31 | 9.94 | 16.88 |
| | +3.88 | +2.86 | +3.50 | +2.35 | +5.86 |
| CausalLM 1.2B | 20.18 | 19.53 | 22.11 | 15.81 | 19.89 |
| ObscuraCoder 1.2B | 28.41 | 30.84 | 33.54 | 22.07 | 31.83 |
| | +8.23 | +11.31 | +11.43 | +6.26 | +11.94 |
| CausalLM 2.8B | 29.78 | 27.99 | 29.19 | 27.11 | 30.07 |
| ObscuraCoder 2.8B | 39.22 | 40.09 | 42.39 | 34.60 | 40.77 |
| | +9.44 | +12.10 | +13.20 | +7.49 | +10.70 |
| StarCoderBase 1B | 23.12 | 23.79 | 24.97 | 18.88 | 22.41 |
| DeepSeekCoder 1.3B | 47.17 | 46.83 | 52.92 | 39.20 | 51.17 |
| StarCoderBase 3B | 30.78 | 33.99 | 36.86 | 30.84 | 34.95 |
| StarCoder2 3B | 50.47 | 47.78 | 51.67 | 48.54 | 49.69 |
| StableCode 3B | 50.81 | 46.89 | 54.64 | 40.17 | 47.12 |
| CodeGemma 2B | 45.31 | 40.07 | 32.63 | 46.11 | 25.77 |
| Phi-2 | 47.77 | 37.13 | 71.78 | 11.76 | 31.38 |

Table 11: Multipl-E `pass@10` comparison between `ObscuraCoder` and comparable Causal LM models, along with frontier models for context.

| Model | C++ | Java | Python | Rust | TypeScript |
|---|---|---|---|---|---|
| CausalLM 255M | 15.97 | 15.22 | 17.08 | 8.93 | 14.62 |
| ObscuraCoder 255M | 16.93 | 18.38 | 25.58 | 11.43 | 18.70 |
| | +0.96 | +3.16 | +8.50 | +2.50 | +4.08 |
| CausalLM 491M | 21.21 | 21.56 | 25.13 | 11.96 | 23.59 |
| ObscuraCoder 491M | 28.53 | 27.88 | 30.04 | 18.91 | 23.96 |
| | +7.32 | +6.32 | +4.91 | +6.95 | +0.37 |
| CausalLM 1.2B | 34.84 | 37.69 | 36.35 | 28.84 | 32.77 |
| ObscuraCoder 1.2B | 48.38 | 51.91 | 49.87 | 40.15 | 48.64 |
| | +13.54 | +14.22 | +13.52 | +11.31 | +15.87 |
| CausalLM 2.8B | 47.71 | 49.44 | 50.12 | 41.42 | 48.98 |
| ObscuraCoder 2.8B | 64.87 | 63.74 | 67.41 | 53.45 | 60.17 |
| | +17.16 | +14.30 | +17.29 | +12.03 | +11.19 |
| StarCoderBase 1B | 39.11 | 38.20 | 40.75 | 30.64 | 40.19 |
| DeepSeekCoder 1.3B | 71.48 | 67.42 | 77.79 | 64.05 | 77.76 |
| StarCoderBase 3B | 55.65 | 56.41 | 61.27 | 55.91 | 58.77 |
| StarCoder2 3B | 69.19 | 67.67 | 76.78 | 77.56 | 73.12 |
| StableCode 3B | 68.34 | 65.48 | 74.81 | 66.21 | 69.55 |
| CodeGemma 2B | 66.17 | 58.85 | 60.77 | 70.27 | 54.63 |
| Phi-2 | 62.46 | 57.40 | 83.95 | 26.91 | 55.76 |

Table 12: Multipl-E `pass@100` comparison between `ObscuraCoder` and comparable Causal LM models, along with frontier models for context.

| ⏱ Model | ⚙ C | ⚙ C++ | 🔵 Go | ☕ Java | 🐍 Python | 🟦 TypeScript | 🦀 Rust |
|---|---|---|---|---|---|---|---|
| CausalLM 255M | 32.07 | 32.45 | 33.13 | 33.79 | 33.09 | 32.19 | 32.58 |
| ObscuraCoder 255M | 32.69 | 33.26 | 34.04 | 34.42 | 33.58 | 32.59 | 33.24 |
| | **+0.62** | **+0.81** | **+0.91** | **+0.63** | **+0.49** | **+0.40** | **+0.66** |
| CausalLM 491M | 33.37 | 33.69 | 35.50 | 34.87 | 34.16 | 33.12 | 33.83 |
| ObscuraCoder 491M | 34.07 | 34.67 | 36.09 | 35.64 | 34.81 | 33.68 | 35.01 |
| | **+0.70** | **+0.98** | **+0.59** | **+0.77** | **+0.65** | **+0.56** | **+1.18** |
| CausalLM 1.2B | 35.28 | 34.97 | 37.19 | 36.61 | 35.32 | 34.59 | 35.59 |
| ObscuraCoder 1.2B | 36.23 | 36.67 | 38.97 | 37.69 | 36.11 | 36.24 | 37.09 |
| | **+0.95** | **+1.70** | **+1.78** | **+1.08** | **+0.79** | **+1.65** | **+1.50** |
| CausalLM 2.8B | 36.09 | 36.41 | 38.76 | 37.38 | 36.14 | 35.40 | 36.86 |
| ObscuraCoder 2.8B | 37.58 | 37.84 | 39.59 | 38.97 | 37.73 | 36.29 | 38.52 |
| | **+1.49** | **+1.43** | **+0.83** | **+1.59** | **+1.59** | **+0.89** | **+1.56** |
| StarCoderBase 1B | 36.11 | 37.02 | 38.75 | 37.40 | 36.88 | 36.07 | 37.13 |
| DeepSeekCoder 1.3B | 35.81 | 35.79 | 35.64 | 36.68 | 35.99 | 35.09 | 36.01 |
| StarCoderBase 3B | 38.69 | 38.49 | 40.03 | 39.08 | 38.14 | 37.59 | 38.52 |
| StarCoder2 3B | 38.09 | 38.78 | 40.48 | 39.14 | 38.39 | 37.45 | 38.89 |
| StableCode 3B | 38.75 | 38.11 | 39.34 | 38.26 | 37.34 | 36.88 | 37.88 |
| CodeGemma 2B | 36.62 | 35.96 | 35.93 | 36.11 | 35.91 | 35.25 | 37.22 |
| Phi-2 | 34.97 | 34.56 | 37.02 | 35.86 | 35.18 | 33.97 | 35.10 |

Table 13: CommitChronicle ROUGE-1 comparison between ObscuraCoder and comparable Causal LM models, along with frontier models for context.

| ⏱ Model | ⚙ C | ⚙ C++ | 🔵 Go | ☕ Java | 🐍 Python | 🟦 TypeScript | 🦀 Rust |
|---|---|---|---|---|---|---|---|
| CausalLM 255M | 9.85 | 9.70 | 10.19 | 11.31 | 10.44 | 9.47 | 10.10 |
| ObscuraCoder 255M | 10.42 | 10.23 | 10.98 | 11.76 | 10.79 | 9.79 | 10.51 |
| | **+0.57** | **+0.53** | **+0.79** | **+0.45** | **+0.35** | **+0.32** | **+0.41** |
| CausalLM 491M | 10.82 | 10.64 | 11.38 | 12.09 | 11.20 | 10.01 | 11.01 |
| ObscuraCoder 491M | 11.22 | 10.92 | 11.83 | 12.51 | 11.56 | 10.59 | 11.45 |
| | **+0.40** | **+0.28** | **+0.45** | **+0.42** | **+0.38** | **+0.58** | **+0.44** |
| CausalLM 1.2B | 11.80 | 11.57 | 12.52 | 13.12 | 12.01 | 10.93 | 11.78 |
| ObscuraCoder 1.2B | 13.16 | 12.95 | 13.98 | 14.11 | 13.08 | 12.04 | 13.08 |
| | **+1.36** | **+1.38** | **+1.46** | **+0.99** | **+1.07** | **+1.11** | **+1.30** |
| CausalLM 2.8B | 13.04 | 12.55 | 13.58 | 14.00 | 12.87 | 11.86 | 12.31 |
| ObscuraCoder 2.8B | 14.28 | 13.97 | 14.91 | 15.24 | 14.14 | 12.41 | 14.12 |
| | **+1.24** | **+1.42** | **+1.33** | **+1.24** | **+1.27** | **+0.55** | **+1.81** |
| StarCoderBase 1B | 12.98 | 12.88 | 14.01 | 14.03 | 13.23 | 12.08 | 13.14 |
| DeepSeekCoder 1.3B | 13.06 | 12.91 | 12.89 | 13.93 | 12.97 | 11.93 | 12.99 |
| StarCoderBase 3B | 14.84 | 14.45 | 14.92 | 15.16 | 14.09 | 13.19 | 14.46 |
| StarCoder2 3B | 14.75 | 14.43 | 15.21 | 15.26 | 14.44 | 13.37 | 14.63 |
| StableCode 3B | 15.46 | 13.87 | 15.05 | 14.65 | 13.87 | 12.89 | 14.63 |
| CodeGemma 2B | 15.08 | 11.84 | 13.47 | 14.06 | 12.77 | 11.86 | 13.96 |
| Phi-2 | 12.18 | 11.77 | 12.19 | 13.13 | 12.38 | 11.19 | 12.28 |

Table 14: CommitChronicle ROUGE-2 comparison between ObscuraCoder and comparable Causal LM models, along with frontier models for context.

| ⏱ Model | ⚙ C | ⚙ C++ | 🔵 Go | ☕ Java | 🐍 Python | 🟦 TypeScript | 🦀 Rust |
|---|---|---|---|---|---|---|---|
| CausalLM 255M | 29.85 | 30.69 | 31.79 | 32.11 | 31.63 | 30.77 | 31.14 |
| ObscuraCoder 255M | 30.47 | 31.52 | 32.07 | 32.74 | 32.11 | 31.23 | 31.79 |
| | **+0.62** | **+0.83** | **+0.28** | **+0.63** | **+0.48** | **+0.46** | **+0.65** |
| CausalLM 491M | 31.21 | 31.99 | 33.39 | 33.22 | 32.75 | 31.64 | 32.62 |
| ObscuraCoder 491M | 31.97 | 32.76 | 34.01 | 33.88 | 33.23 | 32.34 | 33.02 |
| | **+0.76** | **+0.77** | **+0.62** | **+0.66** | **+0.48** | **+0.70** | **+0.40** |
| CausalLM 1.2B | 33.06 | 33.19 | 35.09 | 34.49 | 33.59 | 33.01 | 34.07 |
| ObscuraCoder 1.2B | 34.71 | 34.75 | 36.79 | 35.88 | 35.19 | 34.77 | 35.26 |
| | **+1.65** | **+1.56** | **+1.70** | **+1.39** | **+1.60** | **+1.76** | **+1.19** |
| CausalLM 2.8B | 34.46 | 34.78 | 36.21 | 35.64 | 34.98 | 34.36 | 35.09 |
| ObscuraCoder 2.8B | 36.19 | 36.38 | 37.74 | 37.19 | 36.82 | 35.21 | 36.67 |
| | **+1.73** | **+1.60** | **+1.53** | **+1.55** | **+1.84** | **+0.85** | **+1.58** |
| StarCoderBase 1B | 34.44 | 34.94 | 36.58 | 35.91 | 35.48 | 34.57 | 35.19 |
| DeepSeekCoder 1.3B | 33.94 | 34.42 | 34.24 | 35.40 | 34.61 | 33.82 | 34.52 |
| StarCoderBase 3B | 36.54 | 36.71 | 37.63 | 37.32 | 36.76 | 36.12 | 37.11 |
| StarCoder2 3B | 36.12 | 36.73 | 38.39 | 37.48 | 37.01 | 35.94 | 37.07 |
| StableCode 3B | 37.16 | 36.13 | 37.76 | 36.89 | 36.45 | 35.68 | 36.79 |
| CodeGemma 2B | 35.87 | 34.78 | 35.68 | 35.43 | 35.29 | 34.86 | 35.04 |
| Phi-2 | 33.01 | 32.98 | 34.97 | 34.17 | 34.23 | 32.56 | 33.36 |

Table 15: CommitChronicle ROUGE-L comparison between ObscuraCoder and comparable Causal LM models, along with frontier models for context.

