# OpenReview forum: "ObscuraCoder: Powering Efficient Code LM Pre-Training Via Obfuscation Grounding"
_ICLR.cc/2025/Conference — ICLR 2025 Poster_

### Official Review · Reviewer_P8nX · 2024-10-29

**Soundness:** 3
**Presentation:** 4
**Contribution:** 3
**Rating:** 8
**Confidence:** 4

**Summary:**

This paper introduces ObscuraX and ObscuraCoder, respectively a novel dataset and model for the code LM domain, based on the premise that current approaches to training code language models do not sufficiently take into account the inherent differences in syntactic and semantic structure between programming and natural languages.

Starting from the observation that current data approaches tend to focus on scaling, which may in the near future lead to a data bottleneck, but not on disentangling/making available for learning the syntactic and semantic aspects of modern high-level programming languages, the authors propose a code obfuscation training paradigm, and set out to answer a number of interesting research questions:
1) Does their paradigm lead to improvements on tasks requiring code syntax/semantic understanding?
2) Does obfuscating imports improve library use?
3) Is there any performance regression associated with obfuscation training?
4) How does model performance scale under the proposed training paradigm?

To this end, they lay out in detail how their ObscuraX dataset was created (including code obfuscation, part of which includes obfuscating library import and use), and how their ObscuraCoder models are subsequently trained.\
ObscuraX consists of a large collection of code data scraped from the web; contrary to previous work which uses some code obfuscation as a translation target, the authors introduce a mix of 4 pretraining "objectives": (1) translating original code to obfuscated code; (2) translating obfuscated code to original code; (3) CLM training on original code; (4) CLM training on obfuscated code.

Extensive evaluations against the same model architecture (Llama) on data of comparable size and from the same sources trained as "classic" causal LM (i.e., original code only) indicates that the proposed mixed objective pretraining provides meaningful signals related to syntactic and semantic understanding to the model, improving performance on all evaluated tasks.\
In some conditions, the proposed ObscuraCoder models can even compete with or outperform current frontier models, which are often much larger or trained on at least an order of magnitude more data.\
Comparison to a DOBF-inspired decoder-only model also indicates perfomance gains over using code-to-obfuscation translation as the only added pretraining objective.

**Strengths:**

The paper is very well written and presented. It was very easy to follow, and the authors very clearly laid out the objectives and research questions they want to address.

The paper does an excellent job explaining the data collection and model training methodology, and the provided experiments are meaningful in addressing the posed research questions.

The performance of models trained on the proposed novel obfuscation-grounded dataset indicate that the authors are onto something, and that the proposed training paradigm is meaningful and effective.

**Weaknesses:**

There are some strong claims made about the benefits of the proposed dataset, without testing these by ablating the corpus.\
In particular, RQ2 asks whether obfuscating library imports improved model performance, and is answered in the positive in Section 5; however, this assertion seems to be based on training on data using the full ObscureX obfuscation method, which includes library obfuscation (25%) as well as "standard" code obfuscation.

I am not sure this is a valid conclusion, as these results may as well stem from training on the obfuscated data in general, and not the library obfuscation in particular. It would have been more convincing to use an ablation here and train a model on a version of ObscureX that does not contain library obfuscation at all. This model could then be compared to the model presented here (trained on the full ObscureX data paradigm), to see if there is an improvement that can be specifically attributed to the introduction of library obfuscation.

To some extend this is indicated by the DOBF-based decoder model in Section 6; however, that model is trained differently, with backpropagation on the identifier maps only.

**Questions:**

I really like this paper, and how well-structured and principled it approaches its objective and research questions.

For questions, see my answer to "Weaknesses": Am I right that the shown ablation on DOBF only partially covers RQ2? I don't think it was intended as part of addressing RQ2, since it comes in a section after the main experiments. Did you (or could you?) train models that use ObscureX data that has not gone through library import obfuscation? That would show better if that particular obfuscation method yields additional performance gains past "standard" code obfuscation.

Otherwise, there are just some typos/minor things I noticed during reading:
- Figure 1: The "FUNC_0" on the translation side (purple) of DOBF should be "def"?
- Line 185: "Other work work"
- Line 279: "confiscated code" -> obfuscated?
- Lines 316-317: "[...] as it is nearly an order of magnitude greater that what would be optimal according to the scaling-law." I do not understand this sentence, could you rephrase this?
- Line 518: "effectiveness of out obfuscation" -> our
- Line 522-523: "Unfortunately, we find that post-hoc obfuscation training does bring performance gains to causally pretrained LMs." -> pretty sure this should be "does *not* bring performance gains"?

---

> ### Author Response · Authors · 2024-11-22
> **Official Author Response**
>
> We are deeply encouraged by the reviewer’s positive reception of our work. We address the raised issues below.
>
> > I am not sure this is a valid conclusion, as these results may as well stem from training on the obfuscated data in general, and not the library obfuscation in particular. It would have been more convincing to use an ablation here and train a model on a version of ObscureX that does not contain library obfuscation at all. This model could then be compared to the model presented here (trained on the full ObscureX data paradigm), to see if there is an improvement that can be specifically attributed to the introduction of library obfuscation.
>
> We intended to obfuscate all named entities in the code and started with functions, variables and classes. Owing to resource limitations, we chose to ensure that the few pre-training runs we could pursue to completion would showcase the merits of our idea to its greatest extent. Hence, obfuscating other entities, such as imported modules, was a natural extension given earlier work that has shown the benefits of representing entity spans with single tokens [1].
>
> We acknowledge that our writing could have been more precise on this matter. We clarify that Section 5 RQ2 was only intended to convey the benefits of obfuscation taken to the extreme rather than just the benefits of import obfuscation in isolation.
>
> ------------------------------------------------------------------------------------------------------------------------
> [1] Mohammad Abdul Hadi, Imam Nur Bani Yusuf, Ferdian Thung, Kien Gia Luong, Lingxiao Jiang, Fatemeh H. Fard, David Lo: On the effectiveness of pretrained models for API learning. ICPC 2022: 309-320

---

> > ### Comment · Reviewer_P8nX · 2024-11-25
> >
> > Thank you for your response; as I already think this is a good enough paper, I am keeping my original score.

---

> ### Author Response · Authors · 2024-11-29
> **Thank You**
>
> We thank the reviewer for their feedback and corrections. We would be open to incorporating further suggestions to improve our work.

---

### Official Review · Reviewer_ZqQD · 2024-11-01

**Soundness:** 3
**Presentation:** 3
**Contribution:** 3
**Rating:** 8
**Confidence:** 4

**Summary:**

This paper introduces ObscuraCoder, an approach to enhance the pre-training of Code-LMs by focusing on the disentanglement of syntax and semantics through the use of obfuscated code. Traditionally, Code-LMs rely on autoregressive pre-training which often struggles to handle semantic-preserving syntactic perturbations, thereby reaching a plateau in terms of data efficiency and model robustness. To address these challenges, the authors propose grounding the pre-training on a dataset called ObscuraX, which comprises approximately 55 million pairs of source and obfuscated code across seven programming languages. This method contrasts with previous efforts such as DOBF, which primarily aimed to revert obfuscated code to its original form. ObscuraCoder, ranging from 255M to 2.8B parameters, is pre-trained on a substantial 272B token corpus, including ObscuraX. It achieves significant improvements in syntactic and semantic understanding of code, surpassing both conventional autoregressive pre-training and DOBF models. Notably, ObscuraCoder facilitates enhancements in multilingual code completion, commit summarization, and versatile library-focused code generation, without necessitating modifications to the Code-LM during inference.

**Strengths:**

1. The evaluation of this paper is exceptionally solid:
    1. ObscuraCoder is benchmarked against strong well-established models such as DeepSeekCoder and StarCoder. Those models are very strong at 1B scale, ensuring a competitive comparison.
    2. The model is trained across four different sizes, showing its scalability.
    3. Controlled experiments provide a fair comparison by training ObscuraCoder and baseline models (CausalLM and DOBF) under the same conditions across all scales, attributing its good performance to the proposed methods rather than discrepancies in training setups.
    4. A comprehensive selection of benchmarks is used for evaluation, covering a broad spectrum of tasks across large, multilingual datasets.

2. ObscuraCoder consistently surpasses baseline models in a variety of tasks and scales, with performance gains increasing with model size. This showcases both the effectiveness of the deobfuscation pretraining and its scalability.

**Weaknesses:**

1. The methodology of ObscuraCoder closely resembles the DOBF, which slightly diminishes the novelty of the paper. However, the scope and depth of the experiments and evaluations presented significantly exceed those found in prior DOBF studies.

2. The quality of the pretraining data is not sufficiently controlled in the experiments:
    - The ObscuraX dataset has been curated using Tree-Sitter to drop non-parsable code, and contain more self-contained code. So it contains higher-quality, cleaner code compared to other corpora.
    - Only ObscuraCoder is trained with the higher-quality ObscuraX, whereas the CausalLM and DOBF models are not, raising concerns that performance gains might be attributed to dataset quality rather than the training objective.
    - To verify the effectiveness of the deobfuscation objective, the authors could:
        1. Develop a non-obfuscated version of ObscuraX.
        2. Pretrain a CausalLM using this dataset (mixed with other datasets used by ObscuraCoder, except obfuscated ObscuraX)
        3. Compare its performance against ObscuraCoder to conclusively attribute any improvements to the deobfuscation objective.

3. The paper lacks comprehensive discussion on related works that are highly relevant:
    - While CodeT5 (https://arxiv.org/abs/2109.00859) is mentioned in this paper, CodeT5 actually uses an "identifier-aware" pretraining objective related to code obfuscation. Those pertinent details are not discussed in this paper.
    - AST-T5 (https://arxiv.org/abs/2401.03003), another model trained with syntax-aware objectives and known for its code translation capabilities, should also be included in the related work section.

4. The writing style of the paper, particularly in the abstract and introduction, could be simplified. The use of overly complex vocabulary may hinder understanding of the content.

**Questions:**

N/A

---

> ### Author Response · Authors · 2024-11-22
> **Official Author Response**
>
> We thank the reviewer for their positive reception of our work. We seek to address their remaining concerns below.
>
> > The quality of the pretraining data is not sufficiently controlled in the experiments. The ObscuraX dataset has been curated using Tree-Sitter to drop non-parsable code, and contain more self-contained code. So it contains higher-quality, cleaner code compared to other corpora.
> Only ObscuraCoder is trained with the higher-quality ObscuraX, whereas the CausalLM and DOBF models are not, raising concerns that performance gains might be attributed to dataset quality rather than the training objective. To verify the effectiveness of the deobfuscation objective, the authors could:
> Develop a non-obfuscated version of ObscuraX.
> Pretrain a CausalLM using this dataset (mixed with other datasets used by ObscuraCoder, except obfuscated ObscuraX)
> Compare its performance against ObscuraCoder to conclusively attribute any improvements to the deobfuscation objective.
>
> We point out explicitly in Section 3 how we choose tree-sitter as a parser due to its robustness in the face of wrong or partially sourced code fragments. Our obfuscation pipeline does not drop much in the way of unparseable code in a manner that would make the ObscuraX and the remaining code distribution different. The main source of dropped files during the creation of ObscuraX is timeout errors in our obfuscation pipeline, triggered by extremely long code files. This is unlikely to be the source of significant distribution discrepancy as we largely prune these files from the regular distribution to improve corpus quality [1] (Appendix B.2).
>
> One source of discrepancy that we do concede may be consequential is the overrepresentation of self-contained code in the ObscuraX split. However, our decision to never repeat any sample more than 3 times in either training run should limit the effects of that and place us within the currently acceptable range of repeating data [2].
>
> Finally, we would like to point out that our control versions of the DOBF causal models are also trained on obfuscation splits derived from the ObscuraX dataset.
>
> > The paper lacks comprehensive discussion on related works that are highly relevant:
> While CodeT5 (https://arxiv.org/abs/2109.00859) is mentioned in this paper, CodeT5 actually uses an "identifier-aware" pretraining objective related to code obfuscation. Those pertinent details are not discussed in this paper.
> AST-T5 (https://arxiv.org/abs/2401.03003), another model trained with syntax-aware objectives and known for its code translation capabilities, should also be included in the related work section.
>
> We regret the oversight. We have now cited these highly relevant papers in L180-181.
>
> ------------------------------------------------------------------------------------------------------------------------
> [1] Aaditya K. Singh, Yu Yang, Kushal Tirumala, Mostafa Elhoushi, Ari S. Morcos:
> Brevity is the soul of wit: Pruning long files for code generation. CoRR abs/2407.00434 (2024)
>
> [2] Niklas Muennighoff, Alexander M. Rush, Boaz Barak, Teven Le Scao, Nouamane Tazi, Aleksandra Piktus, Sampo Pyysalo, Thomas Wolf, Colin A. Raffel: Scaling Data-Constrained Language Models. NeurIPS 2023

---

> > ### Comment · Reviewer_ZqQD · 2024-11-25
> >
> > Thank you for your response. I decide to keep my review score of 8.

---

> ### Author Response · Authors · 2024-11-29
> **Thank You**
>
> We thank the reviewer for their feedback and corrections. We would be open to incorporating further suggestions to improve our work.

---

### Official Review · Reviewer_Wout · 2024-11-03

**Soundness:** 3
**Presentation:** 3
**Contribution:** 2
**Rating:** 5
**Confidence:** 4

**Summary:**

This paper proposes a pre-training objective for code language models (Code-LMs) that leverage obfuscated code. The authors claim that "grounding" Code-LMs on obfuscated code helps to disentangle syntax and semantics, which is essential for improving their performance on downstream tasks such as code defect detection, library-oriented code generation, and multilingual code completion.

However, there is a large line of missing literature, mostly on natural language processing and machine learning, that discusses training neural networks to disentangle syntax and semantics in natural language, and some of them even deal with code. In my opinion, the authors should have discussed these papers in more detail and compared their work with them.

Disclosure: my related expertise is more on the machine learning and language modeling side, as well as the related (computational) linguistics perspective. I am not very familiar with the large body of software engineering literature this paper has cited, nor can I evaluate whether the paper has made a significant contribution to the area of SE. However, since ICLR is more considered a machine learning conference, my evaluation is provided from the perspective of machine learning.

**Strengths:**

- The idea is well presented, and the paper is generally easy to follow. All research questions are stated and, to some extent, well answered. A comprehensive ablation study is conducted.
- A promising code model will be released.

**Weaknesses:**

- The authors' background may be more on the software engineering side---most of the cited work is on software engineering, and there is a significant line of literature missing in machine learning and natural language processing.

    In terms of disentangling syntax and semantics, the following two papers discuss training neural networks to disentangle syntax and semantics in natural language:
    - https://aclanthology.org/N19-1254/
    - https://aclanthology.org/P19-1602/

    In terms of semantic-aware language/code generation with language models, the following papers are very relevant:
    - (natural language) https://aclanthology.org/2021.emnlp-main.564/
    - (code) https://aclanthology.org/2022.emnlp-main.231/
    I especially wonder how the performance of the proposed method compares to other models with the decoding strategy proposed in this paper.

    In training code models, the following ICLR 2023 paper is worth comparing to:
    - https://openreview.net/forum?id=hQwb-lbM6EL

    Related to library-oriented code generation, the following ICLR paper is worth referencing:
    - https://openreview.net/forum?id=ZTCxT2t2Ru

- I'm not sure if this comes from the cross-domain difference: to me, with backgrounds in machine learning and linguistics, both syntax and semantics of the original and obfuscated code shown in Figure 1 are all the same. The difference comes from the lexical-level symbols (i.e., variable and function names); therefore, the approach does not really disentangle syntax and semantics but rather disentangling form and meaning. I believe there would be a large amount of ICLR audience who would think the same.

- While the main performance in Table 1 is promising, it is not sufficient to demonstrate the effectiveness of the proposed method. On BigCodeBench, the model performs much worse than most existing models (except StarCode) with a comparable number of parameters Pass@1. Given that DOBF (Lachaux et al., 2021) has already been introduced, I would expect the performance of the proposed model to be better in order to demonstrate enough contribution.

**Questions:**

1. It is not very clear to me what the authors mean by "grounding," which has already been an overloaded term in the machine learning community (cf. Chai et al., 2018). Can the authors provide a more detailed explanation of what they mean by "grounding" in the context? Is this a software engineering term?

[Chai et al., 2018] https://www.ijcai.org/Proceedings/2018/0001.pdf

2. L397: have you tried evaluating Pass@1 with greedy decoding (i.e., temperature = 0)? Since you are anyway submitting one prediction for testing, I would expect greedy decoding generating a program with higher probability (assigned by the model).

3. Table 1: the metrics of ReCode are never introduced in the paper. Please add a brief explanation of what the metrics are.

---

> ### Author Response · Authors · 2024-11-22
> **Official Author Response (Part 1)**
>
> We thank the reviewer for their robust scrutiny of our work. We address the main areas of criticism below.
>
> > The authors' background may be more on the software engineering side---most of the cited work is on software engineering, and there is a significant line of literature missing in machine learning and natural language processing.
>
> Indeed, we acknowledge the deep body of existing work disentangling syntax and semantics in natural language NLP. However, we believe that the structured and grammatically constrained nature of code (small deviations can nullify the correctness of a sample), along with its inherently extensible nature (e.g., the ability to redefine grammar in macros and templates and override meaning with overloading), make Code-LLMs a unique field of research in its own right. In light of this, we focus and take inspiration from existing methods that have shown promise in the sub-field of programming language NLP and have dedicated an entire section of our related work (Section 2 sub-part 1) to “Syntactically and Semantically-Aware Code Representation Learning”. We would appreciate it if the reviewer could list concrete critically relevant ML/NLP references believed to be missing.
>
> > In terms of semantic-aware language/code generation with language models, the following papers are very relevant: https://aclanthology.org/2022.emnlp-main.231/ I especially wonder how the performance of the proposed method compares to other models with the decoding strategy proposed in this paper.
>
> We wish to clarify that our method is purely a training-time intervention on Code-LMs that seeks to more efficiently impart knowledge of code syntax and semantics to the model while leaving its inference-time mode of operation unaltered (autoregressive decoding). Thus, our method does not compete with or preclude any post-hoc decoding-time approach (as is the work presented in the mentioned reference). Our model is thus perfectly amenable to be complemented by recent techniques utilizing test-time-compute and leveraging semantic or execution-aware verifier models [1,2,3] or test-case informed refinements [4,5] in decoding-time for further performance enhancements.
>
> > In training code models, the following ICLR 2023 paper is worth comparing to: https://openreview.net/forum?id=hQwb-lbM6EL. Related to library-oriented code generation, the following ICLR paper is worth referencing: https://openreview.net/forum?id=ZTCxT2t2Ru
>
> We regret our oversight in not citing the hugely influential InCoder [6] paper. The authors are aware of this work and have rectified this mistake in L55. Our choice not to include InCoder in our benchmark comparisons was an informed one dictated by the following considerations:
>
> We conducted a vast array of evaluations across several model families, tasks and settings. This was only made possible by accelerated inference runtimes such as VLLM [7] and SGLang [8], which do not support InCoder. This is also the reason behind the omission of other seminal code LM models, such as CodeGen [9], CodeT5+ [10] and PolyCoder [11], from the evaluations.
>
> The InCoder models are the product of pre-training on a substantially smaller (~52B tokens compared to our ~272B tokens) and less extensively sourced pre-training corpus (no code adjacent data other than StackOverflow) and hence would not represent a fair account of what obfuscation-based training alone brings to the table. Hence, we controlled for all other incidentals and trained our own robust CausalLM baseline model for comparison.
>
> In the interest of alleviating any concerns about omitted models, we urge the reviewer to refer to independently conducted evaluations of CodeGen and CodeT5+ models [12] on popular benchmarks (HumanEval) and compare those to our Python Multipl-E evaluations of ObscuraCoder (Table 10, 11 and 12 in Appendix C). Similarly, the HumanEval performance of InCoder (Table 5) can also be compared to ours. This represents a fair head-to-head comparison. We summarize the salient numbers below showing markedly smaller ObscuraCoder models match or surpass larger formerly-frontier models:
>
> |**Model**       |**Parameters**|**Pass@1**|**Pass@10**|**Pass@100**|
> |----------------|--------------|----------|-----------|------------|
> |**InCoder**     |1.3B          |8.0       |\-         |\-          |
> |**InCoder**     |6.7B          |15.0      |\-         |\-          |
> |**CodeGen-1**   |16B           |18.32     |32.07      |50.80       |
> |**CodeGen-2**   |16B           |20.46     |36.50      |56.71       |
> |**CodeT5+**     |6B            |28.0      |47.11      |69.24       |
> |**ObscuraCoder**|1.2B          |19.79     |33.54      |49.87       |
> |**ObscuraCoder**|2.8B          |26.34 |42.39  |67.41 |
>
> Our reason for excluding the discussion of DocPrompting [13] from our work is due to the RAG-based nature of the approach. While it does target the area of library-oriented code generation, our approach is quite unrelated to this effort.

---

> ### Author Response · Authors · 2024-11-22
> **Official Author Response (Part 2)**
>
> > I'm not sure if this comes from the cross-domain difference: to me, with backgrounds in machine learning and linguistics, both syntax and semantics of the original and obfuscated code shown in Figure 1 are all the same. The difference comes from the lexical-level symbols (i.e., variable and function names); therefore, the approach does not really disentangle syntax and semantics but rather disentangling form and meaning.
>
> We fully agree with the reviewer's contention that by traditional linguistics/NLP parlance, obfuscation is purely a lexical transformation. There is indeed a cross-domain gap here: we follow well established conventions in work on code models that use the term “Syntactic channel” for all surface form aspects of a code’s presentation, including variable, function and struct names [14, 15].
>
> We would further kindly ask  the reviewer to consider the following when evaluating our work:
> Our choice of obfuscating imported modules, as well as platform-level constants, means our transformations are not really syntactically neutral;
>
> It is important to note that our objective provides syntactic information to the model due to our choice to allocate single substitute tokens to obfuscated entities. These spans correspond to whole entities on the syntax tree and have the same effect as the AST-based pre-tokenizer of AST-T5 [16];
>
> Compared to the Causal LM baseline, our approach improves performance on downstream tasks such as Code Defect Detection (Section 5 RQ1) and Code Commit Description (Section 5 RQ3) that demand syntactic understanding of code.
>
> > While the main performance in Table 1 is promising, it is not sufficient to demonstrate the effectiveness of the proposed method. On BigCodeBench, the model performs much worse than most existing models (except StarCode) with a comparable number of parameters Pass@1. Given that DOBF (Lachaux et al., 2021) has already been introduced, I would expect the performance of the proposed model to be better in order to demonstrate enough contribution.
>
> We only include comparisons to frontier models to convey that our methods can be used to train competitive models efficiently. These models are all trained on pre-training corpora **5-20 times larger** than the one we trained on. In many cases, these model releases have important innovations of their own that substantially move the performance needle, such as the synthetic data training of Phi-2 [17] (with its high distribution overlap with benchmarks due to focus on educational code), repo-level pre-training of Deepseekcoder [18] and StarCoder-2 [19] and the multi-stage pre-training for StableCode [20] and CodeGemma [21]. Replicating these innovations would be prohibitively resource-intensive and make it impossible to carry out controlled experiments that isolate the benefits of obfuscation-based pre-training. We hope the reviewer can appreciate that ObscuraCoder models are within reach of several frontier models despite being trained on much smaller corpora and without many of the complex schemes employed in frontier models.

---

> ### Author Response · Authors · 2024-11-22
> **Official Author Response (Part3)**
>
> > L397: have you tried evaluating Pass@1 with greedy decoding (i.e., temperature = 0)? Since you are anyway submitting one prediction for testing, I would expect greedy decoding generating a program with higher probability (assigned by the model).
>
> We have evaluated using the ReCode benchmark (Section 5 RQ1) that already tests models’ abilities on a HumanEval-like task in a greedy decoding setting. In this light, we follow existing work [22, 23] and use low-temperature sampling to evaluate pass@1 and high-temperature sampling to evaluate higher pass rates on HumanEval. In other words, this is a common practice (i.e., established evaluation protocol) for Code-LMs.
>
> > Table 1: the metrics of ReCode are never introduced in the paper. Please add a brief explanation of what the metrics are.
>
> We apologize for the oversight. We have fixed this in L379-381.
>
> ------------------------------------------------------------------------------------------------------------------------
> [1] Freda Shi, et.al.: Natural Language to Code Translation with Execution. EMNLP 2022: 3533-3546
>
> [2] Ansong Ni, et.al.: LEVER: Learning to Verify Language-to-Code Generation with Execution. ICML 2023: 26106-26128
>
> [3] Chenxiao Liu, et. al.: Code Execution with Pre-trained Language Models. ACL (Findings) 2023: 4984-4999
>
> [4] Haau-Sing Li, et.al.: DOCE: Finding the Sweet Spot for Execution-Based Code Generation. CoRR abs/2408.13745 (2024)
>
> [5] Bei Chen, et.al.: CodeT: Code Generation with Generated Tests. ICLR 2023
>
> [6] Daniel Fried, et.al.: InCoder: A Generative Model for Code Infilling and Synthesis. ICLR 2023
>
> [7] https://docs.vllm.ai/en/latest/
>
> [8] https://sgl-project.github.io/
>
> [9] Erik Nijkamp, et.al.: CodeGen: An Open Large Language Model for Code with Multi-Turn Program Synthesis. ICLR 2023
>
> [10] Yue Wang, et.al.: CodeT5+: Open Code Large Language Models for Code Understanding and Generation. EMNLP 2023: 1069-1088
>
> [11] Frank F. Xu, et.al.:
> A systematic evaluation of large language models of code. MAPS@PLDI 2022: 1-10
>
> [12] https://www.salesforce.com/blog/codegen25/
>
> [13] Shuyan Zhou, et.al: DocPrompting: Generating Code by Retrieving the Docs. ICLR 2023
>
> [14] Casey Casalnuovo, et.al.: A theory of dual channel constraints. ICSE (NIER) 2020: 25-28
>
> [15] Santanu Kumar Dash, et.al.: RefiNym: using names to refine types. ESEC/SIGSOFT FSE 2018: 107-117
>
> [16] Linyuan Gong, et.al.: AST-T5: Structure-Aware Pretraining for Code Generation and Understanding. ICML 2024
>
> [17] Phi-2: The surprising power of small language models
>
> [18] Daya Guo, et.al.: DeepSeek-Coder: When the Large Language Model Meets Programming - The Rise of Code Intelligence. CoRR abs/2401.14196 (2024)
>
> [19] Anton Lozhkov, et al.: StarCoder 2 and The Stack v2: The Next Generation. CoRR abs/2402.19173 (2024)
>
> [20] Nikhil Pinnaparaju, et.al.: Stable Code Technical Report. CoRR abs/2404.01226 (2024)
>
> [21] Heri Zhao, et. al.:CodeGemma: Open Code Models Based on Gemma. CoRR abs/2406.11409 (2024)
>
> [22] Raymond Li, et. al.: StarCoder: may the source be with you! Trans. Mach. Learn. Res. 2023 (2023)
>
> [23] Erik Nijkamp, et.al.:CodeGen2: Lessons for Training LLMs on Programming and Natural Languages. CoRR abs/2305.02309 (2023)

---

> ### Comment · Reviewer_Wout · 2024-11-25
> **Thank you for your response.**
>
> I would like to thank the authors for their revision and clarification and their willingness to include the related work. I can confirm. Below, I list and respond to the topics under discussion.
>
> - Syntax and semantics: the points in the authors' response that "it is important to note that our objective provides syntactic information to the model due to our choice to allocate single substitute tokens to obfuscated entities," if I understood correctly, should be translated to "the data synthesis method is syntactically **informed**," which is clearly different from and much weaker than disentangling syntax and semantics.
>
>   I have also double-checked with experts in software engineering and received confirmation from them that obfuscation is not a widely accepted way to disentangle syntax and semantics.
>
> - Baseline models are trained on larger datasets. I understand the authors' point that "these models are all trained on pre-training corpora 5-20 times larger than the one we trained on," however, I wonder if any obstacles are preventing the authors from getting the same amount of data for a completely fair comparison. Or is it possible that you still get similar results even if you train on a comparable amount of data? Considering that the goal of this work is to build a better code LLM, there is no point in making a new mediocre small model using a technique with limited novelty---the current results cannot well support that the proposed method should be adopted.
>
> Could the authors please clarify the usage of "grounding"? The name looks quite confusing. I raised this question in my original review but didn't receive a response.
>
> That being said, the current rating of 3 might be on the harsh side, and I might be able to increase it to 5. However, I still strongly believe the contribution of this paper shouldn't make it into a top-tier machine learning conference like ICLR. I confirm this evaluation after seeing the authors' response.

---

> ### Author Response · Authors · 2024-11-29
> **Official Author Response Round 2 (Part 1)**
>
> We thank the reviewer for making the effort to consult experts in the area while evaluating our work. However, we would like to strongly pushback on their evaluation of our work along the stated lines:
>
> > Syntax and semantics: the points in the authors' response that "it is important to note that our objective provides syntactic information to the model due to our choice to allocate single substitute tokens to obfuscated entities," if I understood correctly, should be translated to "the data synthesis method is syntactically informed," which is clearly different from and much weaker than disentangling syntax and semantics.
> I have also double-checked with experts in software engineering and received confirmation from them that obfuscation is not a widely accepted way to disentangle syntax and semantics.
>
> While there is an ongoing debate on the conventional jargon [1,2] w.r.t how the term syntax may be used w.r.t. describing the surface form presentation of code, we believe that the kind of transformation obfuscation (and by proxy the ObscuraX dataset) represents does not underpin our contribution. Instead, we request that the reviewer take a step back and consider the kind of tasks our objectives present to the model and the specific facets of code understanding they facilitate, as this is ultimately the most important aspect of our work. In the bottom right of Figure 1, we detail the four objectives interspersed throughout the pre-training corpus of ObscuraCoder. They all play a part in making ObscuraCoder models more capable and robust on a per-pretraining-token and per-parameter basis than regular Causal LM models:
>
> 1. The regular CausalLM objective on source code mainly allows the model to grasp the structure of code along with regularly used coding styles and naming conventions. Clearly, It is essential as it represents the usual mode of operation during inference time. While in theory, this objective ought to be syntactically and semantically neutral, in practice, models trained using only CausalLM have been shown to latch on to surface form patterns (their relation to syntax in the linguistic sense notwithstanding) in source code at the cost of semantic understanding [3,4].
>
> 2. The CausalLM objective on obfuscated code is relevant due to its masking of syntactically coherent spans in code, which has been shown to improve the syntactic correctness of the generated code [5].
>
> 3. Source code obfuscation, i.e. source code to obfuscated code translation - same as 2 but with more explicit grounding (more on this later) in source code.
>
> 4. Code deobfuscation, i.e. obfuscated code to source code translation. While there may be field-specific disagreements w.r.t. the nature of obfuscation as a transformation, there is a broad consensus that the task of deobfuscation demands a semantic understanding of code [6,7,8,9,10] when grounded in source code with semantically-derived named entities, which is usually the norm [11]. This provides us with benefits the first three objectives cannot offer by reducing the model’s focus on surface form patterns and demanding syntactic reasoning from the model.
>
> Our claims of disentangling syntax and semantics in Section 1 stem from the combined benefits of these four objectives (especially the last three) during various parts of the ObscuraCode pre-training process in a manner that existing work (including DOBF [12]) does not. We hope that our contributions along these lines are presented in a clearer light now.

---

> ### Author Response · Authors · 2024-11-29
> **Official Author Response Round 2 (Part 2)**
>
> > Baseline models are trained on larger datasets. I understand the author's point that "these models are all trained on pre-training corpora 5-20 times larger than the one we trained on," however, I wonder if any obstacles are preventing the authors from getting the same amount of data for a completely fair comparison. Or is it possible that you still get similar results even if you train on a comparable amount of data? Considering that the goal of this work is to build a better code LLM, there is no point in making a new mediocre small model using a technique with limited novelty---the current results cannot well support that the proposed method should be adopted.
>
> Firstly, we would like to clarify some technical details:
> There are no limits on acquiring obfuscated data other than the availability of unobfuscated source code on the internet. As stated in our response to reviewer ZqQD, we drop merely less than two per cent of files we attempt to obfuscate (mainly due to timeout errors owing to their length).
> Our experiments are primarily limited by our access to compute and the need to additionally pre-train control models (DOBF and CausalLM) for a fair comparison. While our pre-training corpus is smaller than those used by frontier models, we still train on data that is 1-2 orders of magnitude larger than what scaling laws deem optimal [13]. Hence, we focused our available GPU hours on training models at multiple scales to demonstrate that our idea holds up at various scales (Section 5 RQ4).
>
> Secondly, our contribution cannot be distilled into a purely leaderboard-chasing effort but is better viewed in the context of the data-constrained nature of Code LLM development today. Given that the largest open-source code corpora [13,14] are all under a trillion tokens, all recent frontier releases have resorted to repeating training data in pre-training, which has diminishing returns [15,16] and leads to memorization [17, 18]. While this can be partially circumvented by continually pre-training general-purpose LMs [19, 20], we are still data-constrained during the crucial “annealing” phase [21] of pre-training. The capability ceiling this imposes cannot be circumvented even with recent innovations in scaling inference-time compute [22].
>
> The most consequential takeaway of our work is that repeating training data with augmented views rather than as-is improves its effectiveness. This has been demonstrated before during finetuning Code LMs [23] but is especially important in pre-training and represents an actionable insight subsequent frontier model developers can easily incorporate for improved performance and robustness.

---

> ### Author Response · Authors · 2024-11-29
> **Official Author Response Round 2 (Part 3)**
>
> > Could the authors please clarify the usage of "grounding"? The name looks quite confusing. I raised this question in my original review but didn't receive a response.
>
> Indeed, we have not used this term in our work in its originally-coined sense in Cognitive Science but rather in its NLP sense [24] of using augmented data views to impart new knowledge or make existing information more efficiently learnable. We would be open to suggestions of better terms or phrases to express this.
>
> -----------------------------------------------------------------------------------------------------------------------
> [1] Casey Casalnuovo, et. al.: A theory of dual channel constraints. ICSE (NIER) 2020: 25-28
>
> [2] Santanu Kumar Dash, et. al.: RefiNym: using names to refine types. ESEC/SIGSOFT FSE 2018: 107-117
>
> [3] Md. Rafiqul Islam Rabin, Nghi D. Q. Bui, Ke Wang, Yijun Yu, Lingxiao Jiang, Mohammad Amin Alipour: On the generalizability of Neural Program Models with respect to semantic-preserving program transformations. Inf. Softw. Technol. 135: 106552 (2021)
>
> [4] Shounak Naik, et. al.: Probing Semantic Grounding in Language Models of Code with Representational Similarity Analysis. ADMA (2) 2022: 395-406
> [5] Linyuan Gong, et. al.: AST-T5: Structure-Aware Pretraining for Code Generation and Understanding. ICML 2024
>
> [6] ​​Ramtine Tofighi-Shirazi, et. al.: DoSE: Deobfuscation based on Semantic Equivalence. SSPREW@ACSAC 2018: 1:1-1:12
> [7] https://insights.sei.cmu.edu/blog/semantic-code-analysis-for-malware-code-deobfuscation/
>
> [8] Tim Blazytko, et. al.: Syntia: Synthesizing the Semantics of Obfuscated Code. USENIX Security Symposium 2017: 643-659
>
> [9] Chunlin Xiong, Zhenyuan Li, et. al.: Generic, efficient, and effective deobfuscation and semantic-aware attack detection for PowerShell scripts. Frontiers Inf. Technol. Electron. Eng. 23(3): 361-381 (2022)
>
> [10] Sharath K. Udupa, et. al.: Deobfuscation: Reverse Engineering Obfuscated Code. WCRE 2005: 45-54
>
> [11] Earl T. Barr, Premkumar T. Devanbu: The naturalness of software ☆. Perspectives on Data Science for Software Engineering 2016: 51-55
>
> [12] Marie-Anne Lachaux, et. al.: DOBF: A Deobfuscation Pre-Training Objective for Programming Languages. NeurIPS 2021: 14967-14979
>
> [13] https://opencoder-llm.github.io/
>
> [14] Anton Lozhkov, et al.: StarCoder 2 and The Stack v2: The Next Generation. CoRR abs/2402.19173 (2024)
>
> [15] Niklas Muennighoff, et. al.: Scaling Data-Constrained Language Models. NeurIPS 2023
>
> [16] Sachin Goyal, et. al.: Scaling Laws for Data Filtering - Data Curation Cannot be Compute Agnostic. CVPR 2024: 22702-22711
>
> [17] Katherine Lee, et. al.i: Deduplicating Training Data Makes Language Models Better. ACL (1) 2022: 8424-8445
>
> [18] Nicholas Carlini, et. al.: Quantifying Memorization Across Neural Language Models. ICLR 2023
>
> [19] Nikhil Pinnaparaju, et. al.: Stable Code Technical Report. CoRR abs/2404.01226 (2024)
>
> [20] Heri Zhao, et. al.: CodeGemma: Open Code Models Based on Gemma. CoRR abs/2406.11409 (2024)
>
> [21] Cody Blakeney, at. al.: Does your data spark joy? Performance gains from domain upsampling at the end of training. CoRR abs/2406.03476 (2024)
>
> [22] Benedikt Stroebl, at. al.: Inference Scaling FLaws: The Limits of LLM Resampling with Imperfect Verifiers. CoRR abs/2411.17501 (2024)
>
> [23] Indraneil Paul, at. al.: IRCoder: Intermediate Representations Make Language Models Robust Multilingual Code Generators. ACL (1) 2024: 15023-15041
>
> [24] Khyathi Raghavi Chandu, Yonatan Bisk, Alan W. Black: Grounding 'Grounding' in NLP. ACL/IJCNLP (Findings) 2021: 4283-4305

---

> > ### Comment · Reviewer_Wout · 2024-11-29
> >
> > I wanted to thank the authors for their further response. I would like to clarify in response.
> >
> > - Disentangle syntax and semantics: the training objective neither theoretically implies the causal disentanglement between syntax and semantics nor empirically (statistically significantly) outperforms existing models (Table 1); therefore, I can't fully accept this is a strong contribution, as the authors claimed.
> >
> > - Same thing for the leaderboard-chasing arguments. I am clearly not arguing for nor suggesting leaderboard chasing---for work with strong theoretical justification or clear creativity that leads to deeper insights, and it's okay to underperform SotA or even to include no experiments; however, for work with limited technical novelty, it should indeed contribute more to advancing state of the art to get accepted. This paper falls into the latter category.
> >
> > - Grounding: [24] is not a comprehensive review of grounding and is somewhat misleading, as it confuses semantic grounding and communicative grounding. I strongly recommend the authors consult Chai et al. (2018) for the two senses of grounding and re-consider the usage of "grounding" in this paper.
> >
> > [Chai et al., 2018] https://www.ijcai.org/Proceedings/2018/0001.pdf
> >
> > I decided to maintain my rating; however, I acknowledge that the paper is worth some rating around 4, for which I don't quite have the option.

---

> ### Author Response · Authors · 2024-12-02
> **Official Author Response Round 3 (Part 1)**
>
> Thank you for taking the time to respond. We would like to respond to the reviewer’s stated rubrics for evaluating our work as follows:
>
> > Disentangle syntax and semantics: the training objective neither theoretically implies the causal disentanglement between syntax and semantics nor empirically (statistically significantly) outperforms existing models (Table 1); therefore, I can't fully accept this is a strong contribution, as the authors claimed.
>
> To begin with, we would like to point out that we never claim causal disentanglement anywhere in the paper, and this would be an unreasonable bar to impose on us by default, given that implicit disentanglement using a combination of multi-task losses (in our case the four losses in figure 1) is an accepted way of disentangling representations [1] (Section 6.2 of the Review). Such an approach to improve robustness along an axis of variation (in our case, surface form syntax) has been leveraged by a substantial body of prior work at top venues in both Vision [2, 3, 4] and NLP [5, 6] research.
>
> Secondly, we would like to respectfully disagree with the reviewer’s contention that our results on syntactic and semantic knowledge testing tasks in Table 1 aren’t strong. Though the margins are tight on the Code Defect Detection task, our best models beat all but one of the frontier models (StarCoder2-3B). On the ReCode task, our models comfortably beat the StarCoder-1 and the CodeGemma models with substantially fewer pre-training tokens and without using many recent tricks (Mentioned in our Turn 1 response that would’ve made it impossible to isolate the effect of obfuscation-based objectives). Additionally, our models are always significantly better than our Causal LM controlled experiments in every size class, and the aforementioned advantages in robustness hold across a vast array of syntactic perturbations (Tables 7, 8, and 9 in Appendix C).
>
> > Same thing for the leaderboard-chasing arguments. I am clearly not arguing for nor suggesting leaderboard chasing---for work with strong theoretical justification or clear creativity that leads to deeper insights, and it's okay to underperform SotA or even to include no experiments; however, for work with limited technical novelty, it should indeed contribute more to advancing state of the art to get accepted. This paper falls into the latter category.
>
> Our ObscuraCoder models have outperformed Causal LM control models across the board, thus showing the benefits of obfuscation training (RQ 1, 2, and 3) while demonstrating that it doesn't interfere with the regular mode of operation (RQ 4). The evaluation bar that you have detailed while evaluating our work would require anyone proposing a deviation from the existing training approach (including us) to train models on trillions of tokens using a combination of all the latest tricks, such as synthetic test case execution-based filtering [7], repo-level pre-training [8], extended context finetuning [9] and synthetic data pre-training [10]. Given that the pre-training corpora of [7, 8, 10] are not even public, and most frontier models use disparate subsets of these strategies, had we invested our GPU hours into one large pre-training run with all the features turned on, most reviewers (in our opinion rightly) would have opined that our experiments were not sufficiently controlled (impossible to isolate the effects of multi-task obfuscation-based pre-training) and we think most of the ICLR community would concur with us on this point.
>
> Additionally, we would like to point out that comparisons against a tightly controlled baseline, without a view of frontier model performance, is a common practice for new Code LM pre-training advancements, and we believe that penalising our work for contextualizing itself and including comparisons to existing frontier models is quite harsh. Several techniques now commonly incorporated by most CodeLM pre-training runs were initially demonstrated in tightly controlled settings on small corpora without direct comparisons to frontier models. Notable examples include rotary position embeddings [11] (approx. 3.3B token runs), fill-in-the-middle pre-training [12] (100B token runs), perplexity-based pre-training corpora filtering [13] (approx. 33B token runs), and corpus-level semantic de-duplication [14] (26B token runs). We have proven our approach over multiple model sizes and across most major programming languages using substantially larger pre-training jobs (approx. 272-280B token runs). Furthermore, our commitment to releasing the ObscuraX dataset (approx. 119B tokens and 7 languages) along with the multilingual obfuscator code means that our methods can be readily adopted by future frontier model releases. We hope we have answered your doubts to your liking and that you would re-consider your admittedly harsh evaluation of our work in the context of the aforementioned realities.

---

> ### Author Response · Authors · 2024-12-02
> **Official Author Response Round 3 (Part 2)**
>
> Finally, on the issue of grounding:
>
> > Grounding: [24] is not a comprehensive review of grounding and is somewhat misleading, as it confuses semantic grounding and communicative grounding. I strongly recommend the authors consult Chai et al. (2018) for the two senses of grounding and re-consider the usage of "grounding" in this paper.
>
> We thank the reviewer for the illuminating reference [15] to semantic and communicative grounding. Admittedly, as an overloaded term, our NLP-typical usage of the term grounding adheres to semantic grounding. This seems to be the convention followed by existing work in industry [19] , Vision research [16], NLP research [17] and specifically Code LM development [18]. While, we believe our usage of the term keeps with existing practice in LM development, we would be willing to change the term to something more specific like Semantic Grounding, in the camera ready version of the work. We also want to clarify that we are open to particular replacement suggestions of terms that better describe what we are doing.
>
> -----------------------------------------------------------------------------------------------------------------------
> [1] Xin Wang, et. al.: Disentangled Representation Learning. IEEE Trans. Pattern Anal. Mach. Intell. 46(12): 9677-9696 (2024)
>
> [2] Mengxi Jia, et. al.: Learning Disentangled Representation Implicitly Via Transformer for Occluded Person Re-Identification. IEEE Trans. Multim. 25: 1294-1305 (2023)
>
> [3] Taihong Xiao, et. al.: DNA-GAN: Learning Disentangled Representations from Multi-Attribute Images. ICLR (Workshop) 2018
>
> [4] Qi Han, et. al.: RevColV2: Exploring Disentangled Representations in Masked Image Modeling. NeurIPS 2023
>
> [5] Xiongyi Zhang, et. al.: Disentangling Representations of Text by Masking Transformers. EMNLP (1) 2021: 778-791
>
> [6] Hao Zheng, et. al.: Disentangled Sequence to Sequence Learning for Compositional Generalization. ACL (1) 2022: 4256-4268
>
> [7] Binyuan Hui, et. al.: Qwen2.5-Coder Technical Report. CoRR abs/2409.12186 (2024)
>
> [8] Daya Guo, et. al.: DeepSeek-Coder: When the Large Language Model Meets Programming - The Rise of Code Intelligence. CoRR abs/2401.14196 (2024)
>
> [9] Anton Lozhkov, et al.: StarCoder 2 and The Stack v2: The Next Generation. CoRR abs/2402.19173 (2024)
>
> [10] https://www.microsoft.com/en-us/research/blog/phi-2-the-surprising-power-of-small-language-models/
>
> [11] Jianlin Su, et. al.: RoFormer: Enhanced transformer with Rotary Position Embedding. Neurocomputing 568: 127063 (2024)
>
> [12] Mohammad Bavarian, et. al.: Efficient Training of Language Models to Fill in the Middle. CoRR abs/2207.14255 (2022)
>
> [13] Max Marion, et. al.: When Less is More: Investigating Data Pruning for Pretraining LLMs at Scale. CoRR abs/2309.04564 (2023)
>
> [14] Kushal Tirumala, et. al.: D4: Improving LLM Pretraining via Document De-Duplication and Diversification. NeurIPS 2023
>
> [15] Joyce Y. Chai, et. al.: Language to Action: Towards Interactive Task Learning with Physical Agents. IJCAI 2018: 2-9
>
> [16] Yizhen Zhang, et. al.: Explainable Semantic Space by Grounding Language to Vision with Cross-Modal Contrastive Learning. NeurIPS 2021: 18513-18526
>
> [17] Xiaolong Li, et. al.: Semantic Grounding in Dialogue for Complex Problem Solving. HLT-NAACL 2015: 841-850
>
> [18] Shounak Naik, et. al.: Probing Semantic Grounding in Language Models of Code with Representational Similarity Analysis. ADMA (2) 2022: 395-406
>
> [19] https://cloud.google.com/vertex-ai/generative-ai/docs/grounding/overview

---

> > ### Comment · Reviewer_Wout · 2024-12-02
> >
> > I would like to thank the author for their patient multi-round response. I would like to justify my judgment further.
> >
> > > **Though the margins are tight** on the Code Defect Detection task, our best models beat all but one of the frontier models (StarCoder2-3B)
> >
> > As the authors have acknowledged, the improvement is really marginal (e.g., DeepSeekCoder 65.21 vs. 65.58 this work), and the authors haven't included any statistical significance of these results to make the comparison less meaningful. To be clear, I'm not asking for adding experiments on statistical significance, but the authors' claim that the model beats most frontier models in the rebuttal is invalid from a statistical perspective.
> >
> > On other tasks, the model is just in the middle of the mass.
> >
> > > Comparison to tightly controlled experiments.
> >
> > This should always be encouraged. However, in the current shape of the paper, it's unclear to me how these techniques can be translated into further improvement on state-of-the-art models.
> >
> > The authors' response makes me worried more---I still don't think the community should accept a paper with limited novelty and limited improvement to our top-tier conference. Anyway, in recognition of the authors' efforts, I will raise my rating to 5, but I encourage my fellow reviewers and AC to view this line of debate.

---

### Official Review · Reviewer_v87f · 2024-11-03

**Soundness:** 3
**Presentation:** 3
**Contribution:** 3
**Rating:** 6
**Confidence:** 3

**Summary:**

This paper introduces a pre-training strategy that relies on code obfuscation to help models disentangle the semantics and syntax of code during pre-training. Obfuscation is applied to function names, variable names, import statements as well as package names. Some of the details were unclear (especially on the masking applied) but it appears that, given a source code snippet the pretraining data may see the source code as-is (as in regular CausalLM), the source code with function name obfuscation or pre-training with bi-directional obfuscation translation data. In contrast, DOBF a baseline method of obfuscation uses a obfuscated code along with identify maps or regular source-code. Data used for pre-training includes mixed-medium data (code + text) (90B tokens) as well as source-code only data (152B tokens) in the first phrase along with 30B of randomly-shuffled (code+text). In the second phrase, they use 30B tokens of obfuscated code, 58B tokens of obfuscation translation (bi-directional),  and a randomly shuffle of code+text for 30 B tokens. Models are primed with text only before source code is introducted in pre-training. Evaluations presented are presented at different model scales (255M - 2.8B) all with a fixed data budget of 272B tokens. In contrast to using the regular CausalLM training, they report a gain across all metrics on tasks of defect detection, code completion and library oriented code generation. In contrast to DOBF (which appears to be better than vanilla CausalLM), the method still does better.

Overall a good, well documented paper.

**Strengths:**

- Interesting use of code-obfuscation
- Well constructed experiments to help answer the empirical questions posed
- Dataset shall be released after paper publication

**Weaknesses:**

- While the goal of the work was to demonstrate the benefit of code obfuscation (motivated in trying to separate semantics and syntax to improve code-understanding), it would have been helpful to have at least one baseline from the related work in Lines 147-157. At the moment its unclear how one should act on the findings in the paper or how they compare with other approaches (that share the same motivation but are not obfuscation based).

**Questions:**

1. See Weakness. Would it be possible to study this at the smallest scale if there's training budget (and adequate time) during the rebuttal?
2. The motivation for obfuscating library and names and package names was a bit counter-intuitive to me. Wouldn't we want the model to *learn* different package names and what they do? I note the results presented in 391-409 but I am not sure I'm convinced that specifically this type of obfuscation could explain the gain? To make this claim, wouldn't you need to pretrain without this type of obfuscation present?

---

> ### Author Response · Authors · 2024-11-22
> **Official Author Response**
>
> We thank the reviewer for their insightful feedback. Our responses are below.
>
> > While the goal of the work was to demonstrate the benefit of code obfuscation (motivated in trying to separate semantics and syntax to improve code-understanding), it would have been helpful to have at least one baseline from the related work in Lines 147-157. At the moment its unclear how one should act on the findings in the paper or how they compare with other approaches (that share the same motivation but are not obfuscation based).
>
> One of the maxims of our method is to be minimally intrusive to the modern LM pre-training process. Our experiments in Section 5 RQ3 bear this out, demonstrating that these alternative objectives can be mixed with more conventional objectives during pre-training without any “side effects” or performance regressions. This keeps with existing work [1,2]. The existing work in L147-157 primarily employs architectures like GNNs, which cannot be (easily) adapted to the most common modes of operation of modern Code-LMs, i.e., multi-turn interactions with interspersed code and text.
>
> In the rare cases that these approaches use language models [3], due to the aforementioned ability of how our translation pairs can be mixed in with other data to improve models without side effects, the methods in question can always be combined with our obfuscation-based objective and our experiments from Section 5 (RQ3) suggest no negative interference with disparate objectives.
>
> > The motivation for obfuscating library and names and package names was a bit counter-intuitive to me. Wouldn't we want the model to learn different package names and what they do? I note the results presented in 391-409 but I am not sure I'm convinced that specifically this type of obfuscation could explain the gain?
>
> The idea is that when we add regular causal LM into the pre-training mix, the model already learns package names. The same applies when the obfuscation-based translation pairs are used, as we always have unobfuscated code on one side. Obfuscation is there to help the model predict the library name in the context of its use, e.g., predicting `sklearn.cluster.KMeans` when the code calls use the `cluster_centers_` and `inertia_` output from the `CLASS_0.fit()` function. The usage of the obfuscated name is still unchanged due to the syntactically preserving nature of obfuscation (e.g., `KMeans` -> `CLASS_0`, the `CLASS_0` is still being used exactly as `KMeans` before obfuscation; i.e., the class usage doesn’t change).
>
> Additionally, our choice of replacing named spans with a single token (when translating from source code to obfuscated form) demands that models reason about syntactically relevant spans in code, which has also been shown to play a major part in reducing API-calling errors [4].
>
> ------------------------------------------------------------------------------------------------------------------------
> [1] Yue Wang, Weishi Wang, Shafiq R. Joty, Steven C. H. Hoi: CodeT5: Identifier-aware Unified Pre-trained Encoder-Decoder Models for Code Understanding and Generation. EMNLP (1) 2021: 8696-8708
>
> [2] Mohammad Bavarian, Heewoo Jun, Nikolas Tezak, John Schulman, Christine McLeavey, Jerry Tworek, Mark Chen: Efficient Training of Language Models to Fill in the Middle. CoRR abs/2207.14255 (2022)
>
> [3] Chenxiao Liu, Shuai Lu, Weizhu Chen, Daxin Jiang, Alexey Svyatkovskiy, Shengyu Fu, Neel Sundaresan, Nan Duan:
> Code Execution with Pre-trained Language Models. ACL (Findings) 2023: 4984-4999
>
> [4] Mohammad Abdul Hadi, Imam Nur Bani Yusuf, Ferdian Thung, Kien Gia Luong, Lingxiao Jiang, Fatemeh H. Fard, David Lo: On the effectiveness of pretrained models for API learning. ICPC 2022: 309-320

---

> ### Comment · Reviewer_v87f · 2024-11-26
> **Acknowledgement**
>
> Thank you for the response -- the added context may be useful to include in the paper. I would like to retain my scores.

---

> ### Author Response · Authors · 2024-11-29
> **Thank You**
>
> We would like to thank the reviewer for their time. While we completely respect their decision, we would be receptive to feedback or criteria that would improve our work and the reviewer's estimation of it.

---

### Meta-Review · Area_Chair_SAX8 · 2024-12-21

**Metareview:**

The paper introduces a pre-training strategy for code language models using obfuscated code to focus on semantics over syntax. The authors contributed ObscuraX Dataset, 55M source-obfuscated code pairs across seven languages; as well as ObscuraCoder Models, pre-trained models on 272B tokens, ranging from 255M to 2.8B parameters. The authors reported comprehensive results in improved syntactic/semantic code understanding, multilingual code completion, library-oriented code generation, etc. There are some comments on the comparison of this work to related work e.g. DOBF, the effect of obfuscation on semantics vs syntax learning. Please address these comments in the final paper.

**Additional Comments On Reviewer Discussion:**

n/a

---

### Decision · Program_Chairs · 2025-01-22

Accept (Poster)